# Weak Bisimulation Metric-based Representations for Sparse-Reward Reinforcement Learning

## Abstract

Recent studies have shown that bisimulation metrics possess the superiority of essentially extracting the features related to reinforcement learning tasks. However, limited by strict assumptions and the inherent conflict between metrics and sparse rewards, they suffer from serious representation degeneration and even collapse in sparse reward settings. To tackle the problems, we propose a reward-free weak bisimulation metric-based **S**calable **R**epresentation **L**earning approach (**SRL**). Specifically, we first introduce the weak bisimulation metric, which bypasses the intractable reward difference, instead leveraging a trainable Gaussian distribution to relax the traditional bisimulation metrics. Particularly, the Gaussian noise creates a flexible information margin for the metric optimization, which mitigates potential representation collapse caused by sparse rewards. Additionally, due to its pure distribution internally, the metric potentially mitigates representation degeneration resulting from inconsistent computations under strict assumptions. To tighten the metric, we accordingly consider continuous differences over the transition distribution to enhance the accuracy of the initial transition distribution difference, strengthening the extraction of equivalent task features. We evaluate SRL on challenging DeepMind Control Suite, MetaWorld, and Adroit tasks with sparse rewards. Empirical results demonstrate that SRL significantly outperforms state-of-the-art baselines on various tasks. The source code will be available later.

## 1 Introduction

Deep reinforcement learning (DRL) with visual input usually requires learning a low-dimensional state representation from high-dimensional pixels to serve downstream policy learning (Stooke et al., 2021; Ze et al., 2024). Recent literature has demonstrated that the quality of state representation is crucial to the efficiency and performance of policy learning (Tang et al., 2023; Zheng et al., 2024), and the ideal representation should provide sufficient non-redundant information for DRL decision-making (Liao et al., 2023).

To achieve this, previous work has introduced numerous effective representation schemes in contrastive learning (Liu et al., 2023c), temporal prediction (Machado et al., 2023), and data augmentation (Liu et al., 2023b). However, most existing contrastive and temporal prediction methods are limited to learning vague classifications of structural features (Guo et al., 2023). Though they may achieve fast convergence in specific tasks, they struggle with fine-grained feature control (Gao et al., 2022). Regarding data augmentation-based methods, prior research has not yet clarified which effective features are associated with different transformations (Ma et al., 2024), and some setups face difficulties in handling tasks with large action spaces and complex folded regions. Most importantly, the above methods tend to represent all observable dynamics elements, limited by the task-agnostic representation learning (Yuan et al., 2022; Zhang et al., 2021). As a result, these methods remain inadequate for learning tasks with complex dynamics.

In contrast, representation learning built on bisimulation metrics (Ferns et al., 2004; 2011) has fundamentally shown promise in accurately extracting task-relevant information (Liao et al., 2023). Distinct from the aforementioned methods, they extract equivalent task-relevant features by the behavioral similarity metric between states in terms of rewards and dynamics models, i.e., transition

distribution probability (Ferns et al., 2011), which has attracted much attention for their ability to eliminate all task-irrelevant information (Castro, 2020). Subsequently, related work focuses on overcoming its dynamics accuracy problem, e.g., $\pi$-bisimulation metric (Castro, 2020), and introducing additional elements or reward variance to enhance the robustness of the metric, e.g., PSE (Agarwal et al., 2021) and RAP (Chen & Pan, 2022), making it potentially applicable to large-scale tasks. Nevertheless, bisimulation metric-induced representation learning still suffers from information loss and even collapse problems in sparse reward settings (Liao et al., 2023; Zang et al., 2022). This is because, on the one hand, the idealized metric computations with reward difference significantly deviate from practical distances, which potentially leads to approximation bias and inaccurate behavioral similarity measurement (Zang et al., 2022; Castro et al., 2021); On the other hand, sparse or zero reward signals can cause bisimulation metrics to converge to an intractable zero-fixed point, resulting in representation collapse (Liao et al., 2023).

To tackle these issues, we first introduce a weak bisimulation metric, which excludes the unstable reward differences and instead relaxes the strict bisimulation metric with a trainable Gaussian distribution. Concretely, the weak bisimulation metric fine-tunes the metric distance by using the Gaussian noise associated with state transitions, which creates a flexible information margin for the optimization process, effectively avoiding the potential zero-distance representation between states, i.e., the representation collapse problem directly caused by sparse reward differences (Liao et al., 2023). Besides, since the trainable Gaussian distribution shares the same distribution dimensionality as the transition model in computations, the weak metric can potentially avoid inconsistent computations and further approximation bias within traditional metrics caused by strict assumptions (Zang et al., 2022). Although the metric adopts conservative settings, we theoretically demonstrate that it retains certain favorable properties. In further implementations, to strengthen the metric accordingly, we compute multi-step transition distribution differences on the metric to ensure the accuracy of the initial behavioral similarity by considering continuous similarity. Overall, our approach selectively performs effective relaxation or strengthening in specific aspects, and thus we term it scalable representation learning (SRL).

Finally, we conduct extensive experiments on challenging tasks with sparse rewards, i.e., the DeepMind Control, MetaWorld, and Adroit with large action spaces. Notably, most of these tasks feature sparse rewards, meaning they receive tiny rewards before successfully completing a task (Xu et al., 2024). The experimental results demonstrate that SRL significantly outperforms recent various state-of-the-art baselines across a massive number of complex tasks, as highlighted in the best scores shown in Figure 1. Additionally, ablation studies and visualization experiments (available in Appendix D.1 and Appendix D.2) strongly validate the effectiveness of the approach's components in improving representation and policy performance.

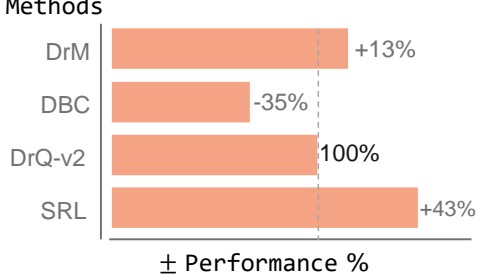

Figure 1: Comprehensive score performance of SRL and baselines compared to DrQ-v2 on DeepMind Control, MetaWorld, and Adroit.

The primary contributions of this work are as follows: (i) We propose a weak bisimulation metric with relaxed properties to address the potential representation instability in bisimulation metrics under sparse reward settings; (ii) Within this metric, we consider continuous transition distribution similarities to strengthen the initial behavior measurement, and then propose a scalable representation learning approach; (iii) We empirically demonstrate that SRL significantly outperforms state-of-the-art baselines on various tasks with sparse rewards, effectively extending the inherent advantages of bisimulation metric-based representations to a wider range of sparse reward scenarios.

## 2 RELATED WORK

DRL with vision as input aims to learn a policy directly from pixel observations (e.g., RGB images), which typically requires increasingly powerful feature representation abilities as the problem scales up (Ze et al., 2023; Nath et al., 2023; Kaufmann et al., 2023; Wang et al., 2023). In earlier work within the Atari 2600 gaming domain, to alleviate the limited representational capacity of end-to-end convolutional encoders, most studies converted RGB images to grayscale (Mnih et al., 2015).

As application scope expanded, recent work has adopted stacked RGB images as input to retain richer detail features (Yarats et al., 2021a; Yuan et al., 2023), but existing representation abilities are struggling to handle these complex and unstructured elements. To address this issue, although self-attention networks (Baee et al., 2021) and Transformers (Parisotto et al., 2020) were introduced early on to enhance feature extraction, Laskin et al. (2020a) demonstrated that these end-to-end optimization methods, sharing DRL gradients, often fail to achieve strong representation performance. As a result, recent work has increasingly focused on constructing effective self-supervised representation losses (Li et al., 2022; Yu et al., 2022), often referred to as auxiliary learning tasks. These approaches typically employ contrastive learning with data augmentation (Liu et al., 2021), latent reconstruction (Yu et al., 2022; Zhou et al., 2024), variational autoencoders (Liu et al., 2022), or the methods based on task elements (e.g., rewards), such as bisimulation metrics (Pavse & Hanna, 2024), reward prediction (Zhou et al., 2023), and state prediction (Fujimoto et al., 2024) to achieve more powerful representations through independent optimization losses.

Previous studies have shown that data augmentation, e.g., rotation, random cropping, random masking, and CycAug (Laskin et al., 2020b) can enhance representation learning algorithms' ability to understand and extract dynamics features (Hansen et al., 2021b; Yarats et al., 2021a). Typically, some research has proposed contrastive learning with data augmentation, where maximizing mutual information between positives and anchors (Oord et al., 2018) enhances consistency across different augmented versions. Nevertheless, Zhang et al. (2021) have pointed out that these methods are often constrained by task-agnostic learning. In other words, due to the randomness of augmentations, these approaches tend to extract all possible dynamics features from observations rather than focus on task-relevant ones (Zhang et al., 2021). To tackle this, some preliminary solutions have been proposed, which introduce actions (Hansen et al., 2021a) or rewards (Yang et al., 2022) to further constrain the dynamics feature set derived from data augmentation, resulting in decision vectors that exclude task-irrelevant features. Although these methods have shown some potential, they tend to underperform in practice, particularly in sparse reward environments.

Recently, the metric-based representation shows how to leverage task-related elements (e.g., rewards) to effectively infer equivalent task information from pixel observations (Liu et al., 2023a). Compared to methods like data augmentation, their key advantage lies in the theoretically task-relevant nature of the extracted features (Zang et al., 2024), such as the bisimulation metric-based approach DBC (Zhang et al., 2021). Specifically, bisimulation metric-based representation learning leverages bisimulation theory, exploiting rewards and transition distribution differences to infer equivalent task information from observations (Castro, 2020). However, numerous studies have shown that rigidly applying the bisimulation is not ideal (Chen & Pan, 2022; Castro et al., 2021), facing two major pitfalls: (i) It is difficult to overcome the computational complexity caused by model inaccuracy (Castro, 2020); (ii) The idealized metric computations with sparse-reward difference significantly deviate from practical distances (Liao et al., 2023). For the former, the $\pi$-bisimulation metric (Castro, 2020) mitigates this issue by considering state similarity under a specific policy. For the latter, while some works suggest relaxing the reward differences within bisimulation by incorporating learned reward variance (Chen & Pan, 2022), successfully applying bisimulation metrics to complex tasks with sparse rewards remains challenging (Liao et al., 2023). Hence, this work seeks to develop a relaxed bisimulation metric to adapt to such complex settings.

## 3 PRELIMINARIES

We start with the underlying assumptions of reinforcement learning (RL) and the notations. Following this, we review the bisimulation metrics used for representation learning, along with the practical optimization challenges.

### 3.1 REINFORCEMENT LEARNING

The interactive environment can be modeled as an infinite-horizon Markov Decision Process (MDP), defined by the tuple $\mathcal{M} = (\mathcal{S}, \mathcal{A}, \mathcal{P}, r, \gamma)$, where $\mathcal{S}$ represents the state space, $\mathcal{A}$ is the continuous action space, $\mathcal{P}(s_{t+1} \mid s_t, a_t) : \mathcal{S} \times \mathcal{A} \times \mathcal{S} \to [0, 1]$ defines the transition distribution that captures the probability from state $s_t$ to state $s_{t+1}$ with action $a_t$, $r : \mathcal{S} \times \mathcal{A} \to \mathbb{R}^1$ denotes the reward function, and $\gamma \in [0, 1)$ is the discount factor. As the image frame exhibits partial observability, we define the state $s_t$ by stacking consecutive image frames. In the scope of representation learning for DRL, a parameterized state encoder $\phi_\omega : \mathcal{S} \to \mathbb{R}^n$ that maps a high-dimensional state to a low-dimensional

vector is learned. Then, the agent's goal is to learn a good encoder and a policy $a_t \sim \pi(\phi_\omega(s_t))$ that maximizes the future cumulative discounted reward $\mathbb{E}_{\mathcal{P},\pi}[\sum_{t=0}^{\infty} \gamma^t r(\phi_\omega(s_t), a_t)]$.

## 3.2 BISIMULATION METRIC

Bisimulation Relation(Givan et al., 2003) (formalized in Appendix B.1) is used to describe the behavioral similarity between abstract states, which is defined as if the rewards they obtain and the probability of the next transition distribution are equal, then they are considered to be behaviorally equivalent(Yarats et al., 2021a). To soften the Bisimulation Relation in the continuous state space (Ferns et al., 2011), past work defines the following pseudo-metric $d : \mathcal{S} \times \mathcal{S} \to \mathbb{R}^1$ to measure the similarity distance between states.

**Theorem 3.1** ($\pi$-bisimulation metric (Castro, 2020)). *Let $d \in \mathbb{M}$ be a pseudometric on $\mathbb{M}$ set over space $\mathcal{S}$. A pseudometric transformation function $\mathcal{F}_B^\pi : \mathbb{M} \to \mathbb{M}$, is defined as,*

$$\mathcal{F}_B^\pi(d)(s_i, s_j) = \left| \mathbb{E}_{a_i \sim \pi} r_{s_i}^{a_i} - \mathbb{E}_{a_j \sim \pi} r_{s_j}^{a_j} \right| + \gamma \mathcal{W}_1(d) \left( \mathcal{P}_{s_i}^\pi, \mathcal{P}_{s_j}^\pi \right) \tag{1}$$

*Where transition model $\mathcal{P}_{s_i}^\pi = \mathbb{E}_{a \sim \pi(s_i)} \mathcal{P}_{s_i}^a$ and $\mathcal{W}_1$ is the 1-Wasserstein distance. $\mathcal{F}_B^\pi$ has a least fixed point $d_B^\pi$ and $d_B^\pi$ is a $\pi$-bisimulation metric.*

**Theorem 3.2** (Value difference bound (Castro, 2020)). *Given any state pair $s_i, s_j \in \mathcal{S}$, and policy $\pi$, we have:*

$$|V^\pi(s_i) - V^\pi(s_j)| \leq d_B^\pi(s_i, s_j) \tag{2}$$

The theorem shows that the value function difference between state pairs is bounded by $d_B^\pi$, which will be used to compare the value difference bound of the weak bisimulation metric.

A zero bisimulation metric value between states indicates that they exhibit similar behavior and state values, yet metric-based representation learning does not aim to find perfectly similar states. Instead, the metric is employed to regularize the encoding distance $\|\phi_\omega(s_i) - \phi_\omega(s_j)\|_2$, so that the pixel states can be clustered to the same distance under latent space as the metric $d_B^\pi(s_i, s_j)$. Typically, the optimization of the bisimulation metric-based representation can be defined as,

$$\mathcal{L}(\phi_\omega) := \mathbb{E}\left[ \left( \|\phi_\omega(s_i) - \phi_\omega(s_j)\|_2 - \left| \mathbb{E}_{a_i \sim \pi} r_{s_i}^{a_i} - \mathbb{E}_{a_j \sim \pi} r_{s_j}^{a_j} \right| - \gamma \mathcal{W}_1(d) \left( \mathcal{P}_{s_i}^\pi, \mathcal{P}_{s_j}^\pi \right) \right)^2 \right] \tag{3}$$

Ideally, minimizing $\mathcal{L}(\phi_\omega)$ loss be able to make $\phi_\omega$ derive task-relevant features in each state and thereby learn the state representations guaranteed by bisimulation metrics. Nevertheless, strict assumptions make optimization challenging. In light of this, Zhang et al. (2021) require a stringent assumption that the states follow a Gaussian distribution, which makes it possible to calculate the closed-form Wasserstein distance with the Euclidean distance, but this is inconsistent with the $L_1$ distance on the reward, resulting in inaccurate bisimulation (Zang et al., 2022). On the other hand, strict bisimulation metrics built on rewards are naturally limited to sparse reward settings, which easily leads to representation collapse (Liao et al., 2023). In this paper, We attempt to improve the strict bisimulation metric and seek a reasonably scalable metric.

## 4 METHODOLOGY

In this section, we first investigate the weaknesses of prior bisimulation metric-based representations in terms of strict assumptions and sparse rewards, as well as the coupling factors that lead to the weaknesses, and then introduce a relaxed bisimulation metric. Then, to strengthen the weak metric accordingly, we implement a strengthened representation learning loss that can more strictly and accurately compute the dynamics transition difference. With these two steps, we finally present a scalable representation learning approach that is particularly suitable for sparse reward settings.

### 4.1 PROBLEMS OF BISIMULATION METRICS

Following the work of Castro (2020), we consider the policy-dependent (on-policy) $\pi$-bisimulation metric $d^\pi$ to optimize the encoder $\phi_\omega$ with parameter $\omega$, where $d^\pi$ can be selected from existing $\pi$-bisimulation-based $d_B^\pi$(Zhang et al., 2021), MICo(Castro et al., 2021), or RAP (Chen & Pan, 2022).

$$\phi^* = \arg\max_{\phi \in \Phi} \mathbb{E}\left[ \left( \|\phi_\omega(s_i) - \phi_\omega(s_j)\|_2 - d^\pi(s_i, s_j) \right)^2 \right] \tag{4}$$

In optional $d^\pi$, since Wasserstein distance is powerful to calculate the distributions' distance, it is often used to calculate internal state transition distribution difference $\mathcal{W}_1(d^\pi)(\mathcal{P}^\pi_{s_i}, \mathcal{P}^\pi_{s_j})$. However, for large-scale tasks with continuous state space, it becomes impractical due to the need to enumerate all possible states. Thus, Zhang et al. (2021) require the strict assumption that the state distribution is a Gaussian, so that the closed-form Wasserstein distance can be calculated with the Euclidean distance in the latent space, which alleviates the computational complexity. However, there is an inconsistency (which can also be understood as a "dimensionality difference") between the $L_1$ distance on rewards and the Euclidean distance, resulting in approximation deviation and further informative degradation (Zang et al., 2022). Although Castro et al. (2021) developed the MICo distance to maintain the consistency of the distance property, this distance that violates the "zero self-distance (Zang et al., 2022)" property may cause the representations to fail. Ultimately, this inconsistency between the transition distribution difference and reward difference makes the above objective (Equation (4)) difficult to optimize.

Besides the problem, the representation performance upon strict bisimulation metrics is unstable when facing sparse reward settings.

**Theorem 4.1** (Liao et al. (2023)). *$\pi$-bisimulation metrics have an upper bound determined by their policy $\pi$ :* $\mathrm{diam}\,(\mathcal{S}; d^\pi) = \sup_{s_i, s_j \in \mathcal{S}} d^\pi(s_i, s_j) \leq \frac{1}{1-\gamma} \sup_{s_i, s_j \in \mathcal{S}} \left| \mathbb{E}_{a_i \sim \pi} r^{a_i}_{s_i} - \mathbb{E}_{a_j \sim \pi} r^{a_j}_{s_j} \right|$.

The proof can be found in the work (Liao et al., 2023). Theorem 4.1 shows that for a specific transition $(s_i, s_j)$, if it yields zero reward signals, the equation $|\mathbb{E}_{a_i \sim \pi} r^{a_i}_{s_i} - \mathbb{E}_{a_j \sim \pi} r^{a_j}_{s_j}| = 0$, which may cause a degenerate solution $\mathrm{diam}(\mathcal{S}; d^\pi) = 0$. Deeply, this indicates that the latent distance between the states converges to a fixed zero, meaning that a collapsed encoder has been learned. For simple dense reward settings, the above problem is more likely to occur in the early training stage, because the initial policy may perform poorly everywhere. Still, from a long-term perspective, the $\pi$-bisimulation metrics will be gradually improved and become an advantage. But seriously, when in sparse reward settings, the reward collected by the policy may be zero everywhere, which may lead to constant representation collapse. Although part of the encoder's training gradients is backpropagated from Temporal-Difference loss in the Critic (Yarats et al., 2021b), this problem will still seriously interfere with the training of $\phi_\omega$.

## 4.2 WEAK BISIMULATION METRIC

We can easily conclude that the internal inconsistency problem of strict bisimulation metrics and the limitation of sparse rewards are actually coupled with the reward signal of the environment. In other words, the reward signal is a fundamental factor that causes the potential performance failure of the metrics. To comprehensively solve the above problems, we first introduce the following weak bisimulation metric to relax typical metrics.

**Definition 4.2** (weak bisimulation metric). *Let $\mathbb{M}$ be the set of all pseudometrics on space $\mathcal{S}$. A pseudometric transformation function $\mathcal{F}^\pi_W : \mathbb{M} \to \mathbb{M}$, is defined as,*

$$\mathcal{F}^\pi_W(d)(s_i, s_j) = \boldsymbol{\epsilon}(s_i, s_j) - \gamma \mathcal{W}_1(d)\left(\mathcal{P}^\pi_{s_i}, \mathcal{P}^\pi_{s_j}\right) \tag{5}$$

*where $\boldsymbol{\epsilon}(s_i, s_j) \sim \mathcal{N}(\mu_c, f_{Var})$ with a constant $\mu_c$ and a nonlinear function $f_{Var}(s_i, s_j; \theta)$ as mean and variance respectively.*

As shown in Definition 4.2, we define the weak bisimulation metric as consisting of the Wasserstein distance of state transition distributions and the parameterized Gaussian distribution $\boldsymbol{\epsilon}(s_i, s_j)$, without the reward difference term.

We emphasize that this is not a semplice combination as it has the following considerations. First, since the reward signal itself is very small under sparse rewards, its contribution to similarity measurements is also weak in a sense. For relaxation reasons, we abandon the consideration of the $|\mathbb{E}_{a_i \sim \pi} r^{a_i}_{s_i} - \mathbb{E}_{a_j \sim \pi} r^{a_j}_{s_j}|$ in Equation (4) and instead introduce a trainable Gaussian distribution $\boldsymbol{\epsilon}(s_i, s_j)$. Specifically, the mean $\mu_{c,(>0)}$ of $\boldsymbol{\epsilon}(s_i, s_j)$ is a tiny constant set according to the sparse reward difference (see Appendix A.2 for details), which aims to potentially provide a non-zero metric to the worst-case $d^\pi(s_i, s_j)$, thereby preventing the degenerate solution $\mathrm{diam}(\mathcal{S}; d^\pi) = 0$ caused by zero reward. In addition, since the variance of $\boldsymbol{\epsilon}(s_i, s_j)$ is determined by the trainable function $f_{Var}$, this also allows the metric to be fine-tuned according to the current transition $(s_i, s_j)$. Finally, we embed the Gaussian distribution $\boldsymbol{\epsilon}(s_i, s_j)$ into a computational dimensionality consistent with

the transition distribution, potentially avoiding the previous inconsistent computational problem between the $L_1$ reward and the Wasserstein distance. This alignment ensures that the overall objective is optimized in the correct approximation direction.

We theoretically analyze some corresponding properties of the weak bisimulation metric below.

**Lemma 4.3** (weak bisimulation metric). *The weak-bisimulation metric is a contraction mapping w.r.t the $L^\infty$ norm on $\mathbb{R}^{S \times S}$, and there exists a fixed-point $d_W^\pi$ for $\mathcal{F}_W^\pi$.*

The proof is available in Appendix A.1. This theory shows that the weak bisimulation metric can converge to a fixed point with iterative learning for any state pairs.

**Theorem 4.4** (Value difference bound). *Given states $s_i$ and $s_j$, and a policy $\pi$, we have,*

$$|V^\pi(s_i) - V^\pi(s_j)| \leq d_W^\pi(s_i, s_j) \tag{6}$$

The proof can be found in Appendix A.2. Theorem 4.4 demonstrates that the difference of the state value function between any state pair is bounded by the weak bisimulation metric $d_W^\pi(s_i, s_j)$.

**Theorem 4.5** (relaxed value difference bound). *Given states $s_i$ and $s_j$, and a policy $\pi$, we have,*

$$d_B^\pi(s_i, s_j) \leq d_W^\pi(s_i, s_j) \tag{7}$$

The proof is provided in Appendix A.2, which shows that $d_W^\pi(s_i, s_j)$ can achieve the difference upper bound of relaxing the original $\pi$-bisimulation metric $d_B^\pi(s_i, s_j)$. It is worth noting that, although it might reduce the correlation between the learned representation and the value function and further introduce redundant information (Chen & Pan, 2022), its advantages could outweigh the drawbacks due to the targeted relaxation for the intractable reward difference term. Specifically, in sparse reward settings, the reward term not only contributes minimally but also risks disrupting the learned representation. For this reason, we conservatively replace the reward term with a Gaussian distribution and set its mean $\mu_c \geq |\mathbb{E}_{a_i \sim \pi} r_{s_i}^{a_i} - \mathbb{E}_{a_j \sim \pi} r_{s_j}^{a_j}|$ (the reward term is typically minimal, details in Appendix A.2), aiming to release only the uncontrollable information measured by the reward difference. Although our metric may introduce certain redundant information, it is at least improvable rather than deadly. Therefore, we still consider this targeted relaxation as a beneficial strategy, with its existing shortcomings set to be addressed in future improvements.

### 4.3 The Scalable Representation Learning

This section analyzes the optimization objective grounded in weak bisimulation and further considers strengthening the objective by accumulating state transition distribution differences accordingly. These two steps ultimately lead to a scalable representation learning loss.

First, according to the weak bisimulation metric (Equation (5)) in Definition 4.2, we set the learning target as:

$$\mathcal{T}(s_i, s_j; \bar{\omega}, \theta) = \epsilon_\theta(\bar{z}_i, \bar{z}_j) + \mathbb{E}_{\substack{s \sim \hat{\mathcal{P}} \\ a \sim \pi}} \gamma \mathcal{W}_1(d_W^\pi) \left( \hat{\mathcal{P}}_\psi^\pi(\bar{z}_i, a), \hat{\mathcal{P}}_\psi^\pi(\bar{z}_j, a) \right) \tag{8}$$

where $z = \phi_\omega(s)$ represents the latent state, $\bar{z}$ is frozen with gradient-free parameter $\bar{\omega}$, and $\psi$ denotes the parameter of the learned dynamics model $\hat{\mathcal{P}}$. The variance of $\epsilon_\theta(\bar{z}_i, \bar{z}_j)$ draws from the network $f_{Var}(\bar{z}_i, \bar{z}_j; \theta)$ with parameters $\theta$. Notably, we train the distribution function $\epsilon_\theta$ by sampling $\tilde{\epsilon} \sim \mathcal{N}(\mu_c, f_{Var})$. To ensure its differentiability, we need to reparameterize $\tilde{\epsilon}$ by, $\tilde{\epsilon} = \mu_c + \sigma_1 f_{Var}(\bar{z}_i, \bar{z}_j; \theta)$, where $\sigma_1 \sim \mathcal{N}(0, 1)$.

Therefore, we give the preliminary representation learning loss based on the target,

$$\mathcal{L}^{weak}(\phi_\omega, \theta) = \mathbb{E}_\mathcal{D} \left[ \|\phi_\omega(s_i) - \phi_\omega(s_j)\|_2 - \mathcal{T}(s_i, s_j; \bar{\omega}, \theta) \right] \tag{9}$$

Although the above objective is effective, it is not enough. As aforementioned, Theorem 4.5 has shown that our metric $d_W^\pi$ has a looser value difference bound than the vanilla bisimulation metric, resulting in introducing potentially redundant information in addition to the superiority of avoiding unstable representations.

To improve this, we try to strengthen the transition distribution difference of the weak bisimulation metric. Specifically, we further come up with the cumulative $\mathcal{W}_1(\hat{\mathcal{P}}_{\bar{z}}^\pi, \hat{\mathcal{P}}_{\bar{z}}^\pi)$ distance of the subsequent $T$-step transition distributions started with $(s_i, s_j)$. The purpose is to make its subsequent

distributions as consistent as possible, thereby improving the distance's accuracy of the initial transition distribution. To balance the accuracy of the parameterized dynamics model $\hat{\mathcal{P}}$, $T$ is actually set to 2, see Section 5.3 for detailed analysis. Notably, since the transition distribution differences are executed over the 50-dimensional latent state, the resulting computational overhead is negligible. As a result, we define the following scalable representation learning loss,

$$\mathcal{L}^{weak}(\phi_\omega, \theta) = \mathbb{E}_\mathcal{D} \left[ \left\| \phi_\omega(s_i) - \phi_\omega(s_j) \right\|_2 - \mathcal{T}^{(T)}(s_i, s_j; \bar{\omega}; \theta) \right]^2 \tag{10}$$

with the learning target,

$$\mathcal{T}^{(T)}(s_i, s_j; \bar{\omega}, \theta) = \boldsymbol{\epsilon}_\theta(\bar{z}_i, \bar{z}_j) + \sum_{t=1}^{t=T} \mathbb{E}_{\substack{s \sim \hat{\mathcal{P}} \\ a \sim \pi}} \left[ \gamma^t \mathcal{W}_1 \left( \hat{\mathcal{P}}_\psi^\pi \left( \bar{z}_i^{(t)}, a^{(t)} \right), \hat{\mathcal{P}}_\psi^\pi \left( \bar{z}_j^{(t)}, a^{(t)} \right) \right) \right] \tag{11}$$

Specifically, due to sparse rewards or zero rewards, methods based on bisimulation metrics are prone to mistakenly infer that two states are bisimilarity based on the one-step transition difference, which causes the encoder to converge prematurely. In fact, a similar problem was also pointed out in the work of Kemertas & Aumentado-Armstrong (2021), which showed that trajectory information may be needed to assist bisimulation metrics under sparse rewards. Given this, we propose the above operation on the weak metric to ensure the consistency of continuous transitions, i.e., latent trajectories, thereby improving the accuracy of current behavior similarity and the extraction of further task-relevant features.

In summary, we propose the above scalable representation learning loss (Equation (10)), where the scalability lies in the purposeful relaxation of the original bisimulation metric and the strict operation of the transfer distribution differentiation, as well as their potential mutual constraints.

### 4.4 OVERALL ARCHITECTURE

We extend the scalable representation learning loss $\mathcal{L}^{weak}(\phi_\omega, \theta)$ built on the weak bisimulation metric to the visual RL algorithm DrQ-v2 (Yarats et al., 2021b), as shown in Figure 2. The encoder $\phi_\omega$ is the core optimization component in the architecture, and its training gradients are propagated by the scalable representation learning loss and Temporal-Difference loss (Lee & He, 2019). Additionally, during the optimization of $\mathcal{L}^{weak}(\phi_\omega, \theta)$, a dynamics model $\hat{\mathcal{P}}_\psi^\pi$ needs to be trained synchronously. Finally, the primary loss denotes $\mathcal{L}^{weak} + \mathcal{L}^\pi + \mathcal{L}^Q$, where $\mathcal{L}^\pi + \mathcal{L}^Q$ is the internal Actor-Critic loss (available in Appendix C.1) of DrQ-v2. The detailed algorithm can be found in Appendix C.3.

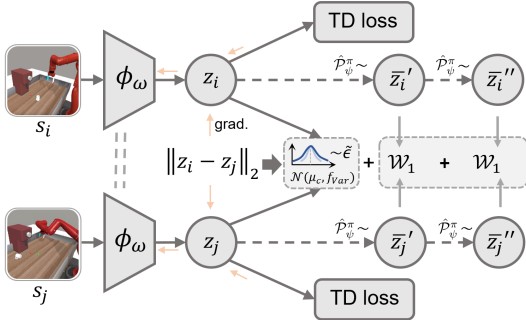

Figure 2: **Overall Architecture.** We use the architecture to describe scalable representation learning for deep reinforcement learning.

## 5 EXPERIMENTS

This section aims to verify SRL's representation ability and policy performance in visual tasks with sparse rewards. We extensively compare SRL with state-of-the-art baselines across three visual environments, ranging from classic physics control to robotic manipulation: DeepMind Control Suite (DMControl) (Tassa et al., 2018), MetaWorld (Yu et al., 2020), and Adroit (Rajeswaran et al., 2017) (depicted in Figure 3). It is worth noting that most of these task settings are highly challenging, featuring rich visual details or large action spaces. Furthermore, the agent can only obtain a larger reward by continuous trial and error until it successfully completes a task.

**Baselines**. We compare against DrQ-v2 (Yarats et al., 2021b), DBC (Zhang et al., 2021), and DrM (Xu et al., 2024) DRL baselines that focus on representation ability or sparse-reward settings. DrQ-v2 is a powerful method based on data augmentation, achieving state-of-the-art sample efficiency on the DMControl benchmark. Of its lesser hyperparameters and stable performance, DrQ-v2 is commonly used as the foundational framework for various DRL methods (Xu et al., 2024). DBC is a bisimulation metric-based representation learning method, where the metric calculates the reward

| (a) DeepMind Control | (b) MetaWorld | (c) Adroit |

Figure 3: Illustration of partial tasks in DMControl, MetaWorld, and Adroit.

and transition distribution difference according to strict assumptions. It shows the superior ability to extract task features in the Distracting DMControl benchmark (Stone et al., 2021). DrM, a state-of-the-art method, proposes a dormant ratio minimization to promote representation and exploration under sparse rewards, achieving the best performance in various environments such as MetaWorld.

**Settings**. We employ the same encoder and hyperparameter settings in the involved methods. Specifically, the encoder consists of a 4-layer convolution network and a 2-layer full connection network with 1024 hidden neurons, which maps three stacked RGB observation images with size $9 \times 84 \times 84$ to a 50-dimensional feature vector for policy learning. Moreover, we utilize the Adam optimizer (Kingma, 2014) with a batch size of 256 and a learning rate of $1e^{-4}$. Distinct from the previous replay buffer with a size of $1e^6$, we set it to the challenging size of $2e^5$. The full hyperparameters are available in Appendix C.2.

### 5.1 DMCONTROL EXPERIMENTS

**DMControl Suite**. DMControl Suite is a widely used benchmark for DRL algorithms, which provides physics control tasks in different difficulties and supports rendering third-person pixel observations as training input (Tassa et al., 2018). We choose complex `walker_run`, `walker_walk`, `reacher_hard` and `quadruped_run` tasks with sparse rewards properties to evaluate our method. The illustrations of DMControl tasks can be seen in Fig. 3. (a).

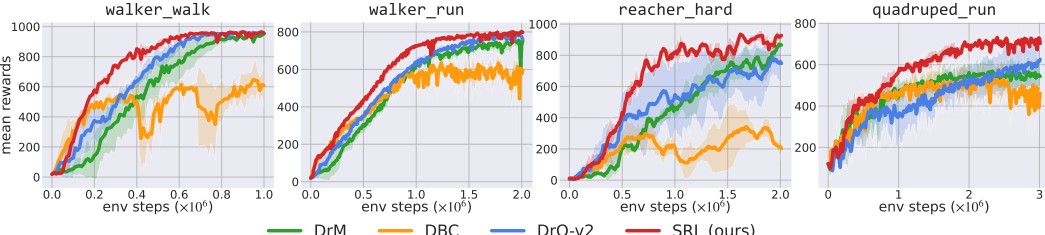

Figure 4: Experimental results on 4 complex tasks in DMControl. Each curve is averaged over three seeds with one standard deviation shaded in the default setting. For each seed, the mean episode reward is evaluated every 5,000 training steps, averaging over 10 episodes.

**DMControl results**. We show the evaluation curves of the SRL method compared with baselines on DMControl in Figure 4. From the upward trend and the convergence range of curves, our method has achieved the best performance in the four tasks, both in terms of convergence speed and best mean episodic rewards. Specifically, with the policy and value network modules kept consistent, the SRL equipped with the weak bisimulation representation learning (red line) significantly outperforms the latest DrM (green line), showing a substantial improvement in the DMControl domain. Notably, compared to the DBC method (yellow line), which is also grounded in bisimulation representation, SRL learns a robust and efficient policy for some challenging tasks, such as the `reacher_hard` task. Overall, we preliminarily conclude that the relaxed weak bisimulation metric-based representation enhances the encoder's ability to extract task-relevant features, providing high-quality latent states for policy learning.

Table 1 records the comparison results on the best mean episode reward. The result shows that SRL has achieved a significant improvement in the best reward compared to baselines. In comparison, the sibling method, i.e., DBC almost achieved the lowest score in the difficult DMControl task, especially the worst performance in `reacher_hard`.

Table 1: Comparison results of the best mean episode reward on complex DMControl tasks.

| Methods | walker_w | walker_r | reacher_h | quad_r |
|---|---|---|---|---|
| DrQ | 944.5±24.6 | 755.2±33.2 | 871.8±30.1 | 570.2±76.9 |
| DBC | 648.4±105.6 | 634.7±38.3 | 342.9±109.5 | 530.9±83.7 |
| DrQ-v2 | 963.5±11.4 | 789.6±19.9 | 784.0±104.6 | 626.4±120.1 |
| SRL (ours) | **965.9±2.0** | **803.2±10.2** | **935.7±12.1** | **732.4±9.9** |

## 5.2 ROBOTIC MANIPULATION EXPERIMENTS

**Robotic Manipulation tasks**. The experiments are evaluated on MetaWorld and Adroit environments with sparse-reward settings, where MetaWorld contains various mechanical arm tasks (e.g., `coffee-push`) and Adroit aims to control a dexterous hand with a large action space to achieve various delicate tasks. The illustrations of MetaWorld and Adroit tasks can be seen in Figure 3 (b) and Figure 3 (c). We emphasize that the field of robotic manipulations is extremely challenging for visual DRL algorithms without real state information. On the one hand, the kinematic structure characteristics of these tasks are extremely complex, requiring the representation algorithm to fundamentally understand the dynamics knowledge and extract the implicit task-relevant features; On the other hand, due to the obvious sparse reward nature, it requires the agent to have long-term planning abilities and remarkable representation performance under sparse rewards.

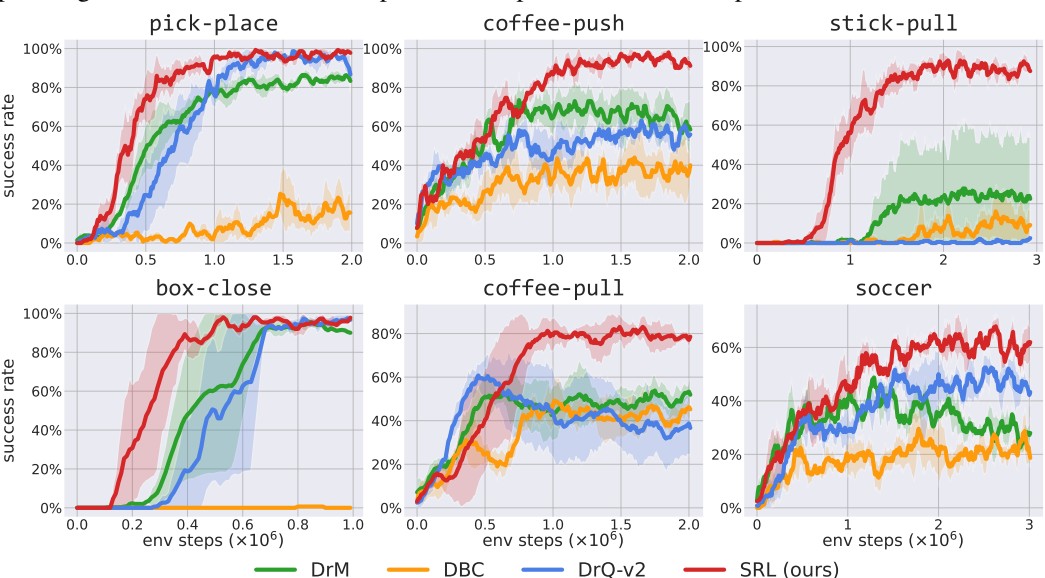

Figure 5: Experimental results on 6 complex tasks with sparse rewards in MetaWorld. Each curve is averaged over three seeds with one standard deviation shaded in the default setting. For each seed, the mean episode reward is evaluated every 5,000 training steps, averaging over 10 episodes.

**MetaWorld results**. As depicted in Figure 5, we evaluate the performance of SRL against baselines on 6 complex tasks with sparse rewards in MetaWorld. Following the experimental details of the work (Xu et al., 2024), we use task success rate as the core comparison metric and train for 1–4 M ($1e^6$) environment steps based on the convergence of tasks. Overall, the SRL method achieves the highest success rate in all tasks and is significantly higher than the suboptimal DrM method. For instance, in the `stick-pull` task, our method achieves an average success rate of nearly 90%, while the baseline struggles to break 30%. In addition, combined with the reward curves in Figure 10 (in Appendix D.3), we can also observe that SRL can quickly obtain rewards and learn policies, and the learning is more stable. In contrast, although the DBC policy has rich task-related elements, it is difficult to obtain reward signals in most tasks, far below the SRL performance. This further empirically shows that, although the bisimulation metric can strictly guarantee that DBC with reward difference achieves equivalent task-relevant information, it is still unable to combine rewards to infer equivalent task features, yet even obtains a damaged representation space, resulting in inefficient policies in these tasks.

**Adroit results**. To verify the visual representation ability of SRL under extremely complex dynamics structures, we also evaluated it on the `Pen` and `Hammer` tasks in the Adroit environment, as shown in Figure 6. Despite the potential blind spots or overlaps of observations rendered from a fixed-position third-person perspective, experimental results show that our

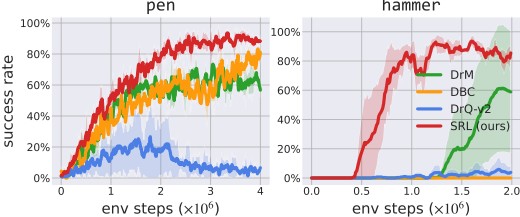

Figure 6: Experimental results on two Adroit tasks with sparse rewards over three seeds.

method can still accelerate training and outperform state-of-the-art methods in these tasks. As for the baseline, we found that DrQ-v2 has difficulty using data augmentation to understand the features of the Adroit task with complex visual structures, which leads to ineffective policy learning. This is consistent with Zhang et al. (2021) and previous experimental results. In addition, although DrM is good at representation learning under sparse rewards, it is still difficult to surpass our method, especially in the `hammer` task.

In Table 2, we record the best mean success rate of SRL and baselines in MetaWorld and Adroit tasks. Our method has a success rate of more than 90% in most tasks and learns an effective policy. In addition, the overall variance of SRL is smaller than that of baselines, so the learned policy is more robust.

Table 2: Comparison results of the best mean success rate (%) on complex MetaWorld and Adroit tasks with sparse rewards.

| Methods | pick-place | coffee-push | stick-pull | box-close | coffee-pull | soccer | pen | hammer |
|---|---|---|---|---|---|---|---|---|
| DrM | 86.4±2.9 | 73.5±14.0 | 27.8±35.8 | 95.7±2.5 | 53.8±7.2 | 48.8±2.0 | 73.3±4.1 | 61.3±43.6 |
| DBC | 25.3±10.9 | 44.7±5.2 | 16.1±13.6 | 0.0±0.0 | 49.2±11.3 | 29.6±8.5 | 83.3±1.9 | 0.0±0.0 |
| DrQ-v2 | 98.7±1.9 | 63.0±6.1 | 2.5±3.5 | 97.5±2.0 | 60.8±5.1 | 52.5±4.4 | 26.7±27.2 | 6.0±8.5 |
| SRL (ours) | **99.3±0.9** | **98.0±0.0** | **92.9±4.6** | **98.0±0.0** | **83.1±6.2** | **67.9±4.8** | **93.5±1.7** | **93.7±3.4** |

## 5.3 ABLATION STUDY

As depicted in Figure 7, to observe how the $T$-step transition distribution differences affect weak bisimulation metric-based representation learning, we execute the ablation study of SRL with the settings of step $T = \{1, 2, 3\}$ in several tasks. To better distinguish the convergence performance among policies, we record the score of 1/2 total training frames. From the figure, we can draw the following results: i) The performance of the SRL policies with $T = \{1, 2, 3\}$ settings all surpasses the DrQ-v2 ($T = 0$), which strongly demonstrates the effectiveness of the fundamental weak bisimulation metric; ii) For the specific $T = \{1, 2, 3\}$, the policies performance is $SRL_{T=2} > SRL_{T=1} > SRL_{T=3}$. In other words, cumulative multi-step transition distribution differences have the potential to improve the representation ability of weak bisimulation metric, but its perfor-

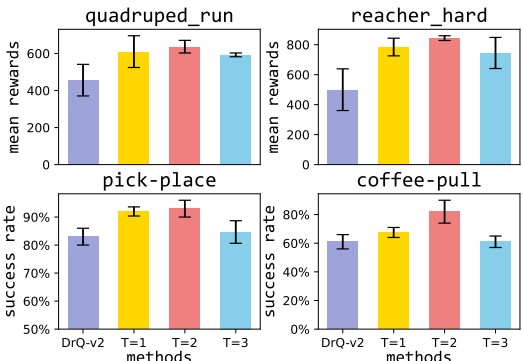

Figure 7: Ablation study in SRL. Each result is the average of three seeds, where the boundary represents the standard deviation. DrQ-v2 can be regarded as a degenerate version with $N = 0$.

mance tends to decay with the increase of the steps. We argue that the decay at $T = 3$ may be led by the accumulated deviation of the learned dynamics model $\hat{\mathcal{P}}_\psi$. For this reason, we set $T = 2$ to tighten the state similarity measure of the weak bisimulation metric in the scalable representation learning loss, achieving the best representation learning performance. Note again that since the transition differences are performed on the latent states, the computational overhead remains negligible.

## 6 CONCLUSION

In this paper, we propose a simple but efficient scalable representation learning approach (SRL) based on a new weak bisimulation metric in sparse reward settings. Overall, SRL alleviates the potential inefficient representations and limitations caused by strict bisimulation metrics under sparse reward settings by relaxing the reward difference term and strengthening the effectiveness of transition distribution differences. Additionally, we briefly analyze the favorable value difference bound and the convergence of the weak bisimulation metric. Extensive comparative experiments across complex tasks in the DMControl, MetaWorld, and Adroit environments verify the state-of-the-art performance of these simple settings. Finally, SRL extends the methods' advantages built on the bisimulation concept, i.e., inherently good at extracting task-relevant information, to a wider range of sparse-reward task scenarios.

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

# Appendices

## A  PROOFS

### A.1  THE PROOF OF LEMMA 4.3

**Lemma 4.3** (weak bisimulation metric)**.** *The weak-bisimulation metric is a contraction mapping w.r.t the $L^\infty$ norm on $\mathbb{R}^{\mathcal{S} \times \mathcal{S}}$, and there exists a fixed-point $d_W^\pi$ for $\mathcal{F}_W^\pi$.*

*Proof.* To prove that the $\mathcal{F}_W^\pi$ function has a fixed point $d_W^\pi$, we first need to prove that it is a contraction mapping.

As Definition 4.2, given the fixed expectation of $\boldsymbol{\epsilon}(s_i, s_j)$, weak-bisimulation metric is,

$$\mathcal{F}_W^\pi(d)(s_i, s_j) = \boldsymbol{\epsilon}(s_i, s_j) - \gamma \mathcal{W}_1(d)\left(\mathcal{P}_{s_i}^\pi, \mathcal{P}_{s_j}^\pi\right) \tag{12}$$

Consider $d,\ d' \in \mathbb{M}$, we derive,

$$\left|\mathcal{F}_W^\pi(d)(s_i, s_j) - \mathcal{F}_W^\pi(d')(s_i, s_j)\right| \tag{13}$$

$$= \boldsymbol{\epsilon}(s_i, s_j) + \mathcal{W}_1(d)\left(\mathcal{P}_{s_i}^\pi, \mathcal{P}_{s_j}^\pi\right) - \boldsymbol{\epsilon}(s_i, s_j) - \mathcal{W}_1(d')\left(\mathcal{P}_{s_i}^\pi, \mathcal{P}_{s_j}^\pi\right)$$

$$= \left|\gamma\left(\mathcal{W}_1(d)\left(\mathcal{P}_{s_i}^\pi, \mathcal{P}_{s_j}^\pi\right) - \mathcal{W}_1(d')\left(\mathcal{P}_{s_i}^\pi, \mathcal{P}_{s_j}^\pi\right)\right)\right|$$

$$= \left|\gamma \sum_{s_i', s_j'} \pi(a_i|s_i)\, \pi(a_j|s_j)\, \mathcal{P}_{s_i}^{a_i}(s_i')\, \mathcal{P}_{s_j}^{a_j}(s_j')\,(d - d')\,(s_i', s_j')\right|$$

$$\leq \gamma \|d - d'\|_\infty.$$

Therefore, $\mathcal{F}_W^\pi(d)$ is a contraction mapping w.r.t. the $L_\infty$ norm and there exists a unique fixed-point $d_W^\pi$ for $\mathcal{F}_W^\pi(d)$. $\qquad\square$

### A.2  PROOFS OF THEOREM 4.4 AND THEOREM 4.5

Before proving Theorem 4.4 and Theorem 4.5, we need to present a weak assumption regarding the reward expectation under sparse reward settings. Then, we set the hyperparameter $\mu_c \geq |\mathbb{E}_{a_i \sim \pi} r_{s_i}^{a_i} - \mathbb{E}_{a_j \sim \pi} r_{s_j}^{a_j}|$, where $\mu_c$ is the mean of the trainable Gaussian distribution $\boldsymbol{\epsilon}(s_i, s_j)$ in the weak bisimulation meric, i.e., $\mu_c = \mathbb{E}_{s_i, s_j \sim \mathcal{P}}[\boldsymbol{\epsilon}(s_i, s_j)]$.

**Assumption A.1** (sparse-reward expectation)**.** *Given a reward function $r_{s_i}^{a_i} = r(s_i, a_i)$ with sparse reward settings, the expectation of $r_{s_i}^{a_i}$ before obtaining the success reward satisfies $\mathbb{E}_{a_t \sim \pi} r_{s_t}^{a_t} \leq C_1$, where $C_1$ is a sufficiently small constant.*

Taking the verification environment of this work with sparse rewards as an example, experience shows that almost all transition rewards are usually minimal or even zero before reaching the goal. Therefore, the above Assumption A.1 is a weak assumption that is easy to satisfy in a sparse reward environment.

Then, based on the above weak assumptions, we can easily obtain,

$$0 \leq \left|\mathbb{E}_{a_i \sim \pi} r_{s_i}^{a_i} - \mathbb{E}_{a_j \sim \pi} r_{s_j}^{a_j}\right| \leq 2C_1. \tag{14}$$

In order to make $|\mathbb{E}_{a_i \sim \pi} r_{s_i}^{a_i} - \mathbb{E}_{a_j \sim \pi} r_{s_j}^{a_j}| \leq \mu_c$ hold, we only need to satisfy $C_1 \leq {}^1\!/{2}\mu_c$ in the actual implementation, where the hyperparameters $C_1$ and $\mu_c$ can be seen in Appendix C.2.

In light of the above conclusions, we then prove Theorem 4.4 and Theorem 4.5 respectively.

**Theorem 4.4** (Value difference bound)**.** *Given states $s_i$ and $s_j$, and a policy $\pi$, we have,*

$$|V^\pi(s_i) - V^\pi(s_j)| \leq d_W^\pi(s_i, s_j) \tag{15}$$

*Proof.* We follow the work (Castro, 2020) to prove the value function difference bound. To prove the above theory, we first introduce the standard value function $V_{n+1}^\pi(s_i) = \mathbb{E}_{a_i \sim \pi} r_{s_i}^{a_i} + \gamma \sum_{s' \in \mathcal{S}} \mathcal{P}_{s_i}^\pi(s') V_n^\pi(s')$ and the update operation with the property of contraction mapping (see Lemma 4.3),

$$d_{W,n+1}^\pi(s_i, s_j) = \boldsymbol{\epsilon}(s_i, s_j) + \gamma \mathcal{W}_1(d_n^\pi)\left(\mathcal{P}_{s_i}^\pi, \mathcal{P}_{s_j}^\pi\right) \tag{16}$$

with initial $V_0^\pi \equiv 0$ and $d_{W,0}^\pi \equiv 0$.

Then, we proceed it by induction. For initial case $n = 0$, obviously $|V_0^\pi(s_i) - V_0^\pi(s_j)| \leq d_{W,0}^\pi(s_i, s_j)$ holds, and we then suppose true up to case $n$. Therefore, we have,

$$\left| V_{n+1}^\pi(s_i) - V_{n+1}^\pi(s_j) \right|$$

$$= \left| \mathbb{E}_{a_i \sim \pi} r_{s_i}^{a_i} + \gamma \sum_{s' \in \mathcal{S}} \mathcal{P}_{s_i}^\pi(s') V_n^\pi(s') - \left( \mathbb{E}_{a_j \sim \pi} r_{s_j}^{a_j} + \gamma \sum_{s' \in \mathcal{S}} \mathcal{P}_{s_j}^\pi(s') V_n^\pi(s') \right) \right|$$

$$\leq \mathbb{E}_{a_i \sim \pi} r_{s_i}^{a_i} - \mathbb{E}_{a_j \sim \pi} r_{s_j}^{a_j} + \left| \gamma \sum_{s' \in \mathcal{S}} V_n^\pi(s') \left( \mathcal{P}_s^\pi(s') - \mathcal{P}_t^\pi(s') \right) \right|$$

$$\leq \left| \mathbb{E}_{a_i \sim \pi} r_{s_i}^{a_i} - \mathbb{E}_{a_j \sim \pi} r_{s_j}^{a_j} \right| + \left| \gamma \mathcal{W}(d_{W,n}^\pi)\left(\mathcal{P}_{s_i}^\pi, \mathcal{P}_{s_j}^\pi\right) \right|$$

$$= \left( \left| \mathbb{E}_{a_i \sim \pi} r_{s_i}^{a_i} - \mathbb{E}_{a_j \sim \pi} r_{s_j}^{a_j} \right| - \boldsymbol{\epsilon}(s_i, s_j) \right) + \left( \boldsymbol{\epsilon}(s_i, s_j) - \left| \gamma \mathcal{W}(d_{W,n}^\pi)\left(\mathcal{P}_{s_i}^\pi, \mathcal{P}_{s_j}^\pi\right) \right| \right)$$

$$\leq \boldsymbol{\epsilon}(s_i, s_j) + \left| \gamma \mathcal{W}(d_n^\pi)\left(\mathcal{P}_{s_i}^\pi, \mathcal{P}_{s_j}^\pi\right) \right| = d_{W,n+1}^\pi(s_i, s_j) \tag{17}$$

where the second inequality comes from the dual representation of the Wasserstein distance (Villani, 2021), and the last inequality is true when satisfying $C_1 \leq {}^1\!/2\mu_c$ based on the Assumption A.1. By the above steps, we can summarize that, $\forall n \in \mathbb{N}, \forall s_i, s_j \in \mathcal{S}, |V_{n+1}^\pi(s_i) - V_{n+1}^\pi(s_j)| \leq d_{W,n+1}^\pi(s_i, s_j)$ holds. $\square$

**Theorem 4.5** (relaxed value difference bound). *Given states $s_i$ and $s_j$, and a policy $\pi$, we have,*

$$d_B^\pi(s_i, s_j) \leq d_W^\pi(s_i, s_j) \tag{18}$$

*Proof.* To prove $d_B^\pi(s_i, s_j) \leq d_W^\pi(s_i, s_j)$, we just need to prove $d_W^\pi(s_i, s_j) - d_B^\pi(s_i, s_j) \geq 0$. Clearly, given $C_1 \leq {}^1\!/2\mu_C$, we have,

$$d_W^\pi(s_i, s_j) - d_B^\pi(s_i, s_j)$$

$$= \left( \gamma \mathcal{W}_1(d)\left(\mathcal{P}_{s_i}^\pi, \mathcal{P}_{s_j}^\pi\right) + \epsilon(s_i, s_j) \right)$$

$$- \left( \left| \mathbb{E}_{a_i \sim \pi} r_{s_i}^{a_i} - \mathbb{E}_{a_j \sim \pi} r_{s_j}^{a_j} \right| + \gamma \mathcal{W}_1(d)\left(\mathcal{P}_{s_i}^\pi, \mathcal{P}_{s_j}^\pi\right) \right)$$

$$= \boldsymbol{\epsilon}(s_i, s_j) - \left| \mathbb{E}_{a_i \sim \pi} r_{s_i}^{a_i} - \mathbb{E}_{a_i \sim \pi} r_{s_i}^{a_i} \right| \geq 0 \tag{19}$$

Then, $d_B^\pi(s_i, s_j) \leq d_W^\pi(s_i, s_j)$ holds. $\square$

# B  BACKGROUND SUPPLEMENT

## B.1  BISIMULATION RELATION

Bisimulation Relations can be applied to group states in Markov Decision Processes (MDPs) that are behaviorally equivalent, aiding in state space reduction and efficient policy learning. Bisimulation Relation is defined as follows.

**Definition B.1** (Bisimulation Relations (Givan et al., 2003)). *Given an MDP $\mathcal{M} = (\mathcal{S}, \mathcal{A}, \mathcal{P}, r)$, a relation $E \subseteq \mathcal{S} \times \mathcal{S}$ is a bisimulation relation, if whenever $(s_i, s_j) \in E$ the following properties hold,*

$$\mathcal{R}(s_i, a) = \mathcal{R}(s_j, a) \quad \forall a \in \mathcal{A} \tag{20}$$

$$\mathcal{P}(G|s_i, a) = \mathcal{P}(G|s_j, a) \quad \forall a \in \mathcal{A}, \quad \forall G \in \mathcal{S}_B \tag{21}$$

*where $\mathcal{S}_E$ is the partition of $\mathcal{S}$ defined by the relation $E$ (the set of all groups $G$ of equivalent states), and $\mathcal{P}(G|s, a) = \sum_{s' \in G} \mathcal{P}(s'|s, a)$.*

## B.2 REPRESENTATIONS IN SPARSE REWARDS

Reward signals can implicitly represent specific or abstract regions in observations that cause rewards (Yarats et al., 2021a). Recently, some work has modeled reward signals to obtain useful representation information from observations, such as bisimulation metric representation based on reward differences (Wang et al., 2024). However, when faced with real tasks with sparse rewards, metrics tend to converge to the zero-fixed point due to continuous zero rewards, i.e., representation collapse (Liao et al., 2023). Although prior work has shown that modeling reward sequences can collect richer reward signals (Kemertas & Aumentado-Armstrong, 2021; Yang et al., 2022), the above problem is still difficult to avoid. Instead, it is more important to balance the dependence on reward utilization in sparse reward environments.

# C IMPLEMENTATIONS

## C.1 ACTOR-CRITIC

Following the work Yarats et al. (2021b), we employ the Actor-Critic (AC) algorithm (Konda & Tsitsiklis, 1999) as the backbone framework of DrQ-v2, where the DrQ-v2 is the basic structure of the above experimental approaches. In general, AC is an off-policy reinforcement learning algorithm for continuous control, consisting of a Critic network with a value function $Q_\vartheta(s_t, a_t)$ and an Actor network with a policy function $\pi_\upsilon(s_t)$. Similar to Barth-Maron et al. (2018), this setting uniformly uses the $n$-step return value to enhance the Temporal-Difference error estimation, and uses double $Q_{k=0}$ and $Q_{k=1}$ to alleviate the overestimation bias. Therefore, we train the Critic network using the following loss:

$$\mathcal{L}^Q\left(\vartheta_k, \phi_\omega\right) = \mathbb{E}_\mathcal{D}\left[Q_{\vartheta_k}\left(\phi_\omega\left(s_t\right), a_t\right) - \left(\sum_{i=0}^{n-1} \gamma^i r_{t+i} + \gamma^n \min_{k=0,1} Q_{\bar\vartheta_k}^{\text{tgt}}\left(\phi\left(s_{t+n}\right), a_{t+n}\right)\right)\right]^2 \tag{22}$$

where $Q_{\bar\vartheta_k}^{\text{tgt}}$ is the target Q function with frozen network parameters $\bar\vartheta_k$, and $\bar\vartheta_k$ is updated from the exponential moving average (EMA) of the trainable parameters $\vartheta_k$. $\mathcal{D}$ represents the experience replay buffer in off-policy DRL.

In addition, since DrQ-v2 adopts a deterministic policy, I train the Actor network with parameter $\upsilon$ with the following loss,

$$\mathcal{L}^\pi(\upsilon) = -\mathbb{E}_\mathcal{D}\left[\min_{k=0,1} Q_{\vartheta_k}\left(\phi_\omega\left(s_t\right), \pi_\upsilon\left(\phi_\omega\left(s_t\right)\right) + \varepsilon\right)\right] \tag{23}$$

where the action noise $\varepsilon \sim \text{clip}(\mathcal{N}(0, \sigma^2))$ is used to ensure the stochastic nature in the deterministic policy. In the interactive phase, $\varepsilon \sim \mathcal{N}(0, \sigma_t^2)$, where $\sigma_t$ is scheduled standard deviation for the exploration noise. Notably, the encoder parameters $\phi_\omega$ will not be trained by the Actor gradient, see DrQ-v2 for details.

## C.2 HYPERPARAMETERS

For the approaches involved in the experimental section, the main encoder network consists of 4 convolutional layers with the same filter sizes $3 \times 3$ and strides 2, 1, 1 and 1, respectively. In the Actor and Critic networks, an encoder trunk network is independently set up to map the encoded convolution output to a 50-dimensional feature vector, thereby serving the learning of policies and value functions. For the settings of the SRL representation module, we set a 2-step transition distribution difference to achieve a trade-off between the utilization of the dynamics model and the deviation caused by its accuracy. In addition, we set the sparse reward expectation $C_1 = 0.1$ in Assumption A.1. To satisfy the condition $C_1 \leq 1/2\mu_c$, we conservatively set the mean $\mu_c = 0.5$ of the Gaussian distribution function $\mathcal{N}(\mu_c, \Sigma_\theta)$. It is worth noting that the replay buffer size in this work is set to $2e^5$, which is 20% of the previous capacity. This can effectively verify the efficiency of approaches while avoiding huge resource consumption. In fact, the experimental results also show that the replay buffer size of $2e^5$ may be sufficient. Please see the table below for detailed hyperparameters.

Table 3: Networks dimensions and hyperparameters.

| Hyperparameter | Shared Setting |
|---|---|
| Training steps | $1 \sim 4\text{M}$ |
| Seed frames | 4000 |
| Exploration steps | 2000 |
| Evaluation episodes | 10 |
| Replay buffer capacity | $2e^5$ |
| Episode length | 1000 for DMControl, 500 for MetaWorld, 200 for `pen` and 100 for `hammer` |
| Batch size | 512 for `walker_run` and `walker_walk`, otherwise 256 |
| Frame stack | 1 for Adroit, otherwise 3 |
| Discount factor $\gamma$ | 0.97 for MetaWorld, otherwise 0.99 |
| State dims | $9 \times 84 \times 84$ |
| Encoder conv kernels | [32,32,32,32] |
| Encoder conv filter size | $[3 \times 3, 3 \times 3, 3 \times 3, 3 \times 3]$ |
| Encoder conv strides | [2,1,1,1] |
| Hidden dims | 1024 |
| Latent state dims | 50 |
| Action repeat | 2 |
| $n$-step returns | 3 |
| Optimizer | Adam |
| Learning rate | $1e^{-4}$ |
| $\tau_Q$ | 0.01 |
| Gradient training frequency | 2 |
| Exploration temperature | 0.1 |
| **Hyperparameter** | **SRL Setting** |
| State transition step $T$ | 2 |
| Sparse reward expectation $C_1$ | 0.1 |
| Trainable Gaussian distribution mean $\mu_c$ | 0.5 |

## C.3 LEARNING ALGORITHM

We describe the main algorithm steps as follows.

---

**Algorithm 1** Scalable Representation Learning (SRL)

---

1: Initialize a replay buffer $\mathcal{D}$ with size $N$, encoder $\phi$, policy $\pi$ , latent model $\hat{\mathcal{P}}_\psi$.
2: **for** $m \leftarrow 1$ to (# epochs) **do**
3:    **for** $i \leftarrow 1$ to (# episodes per epoch) **do**
4:       Encode state $z_t = \phi_\omega(s_t)$
5:       Execute action $a_t \sim \pi_\phi(z_t) + \varepsilon$ where $\varepsilon \sim \mathcal{N}(0, \sigma_t^2)$.
6:       Run a step in environments $s_{t+1} \sim \mathcal{P}(\cdot|s_t, a_t)$
7:       Collect data $\mathcal{D} \leftarrow \mathcal{D} \cup \{s_t, a_t, r_{t+1}, s_{t+1}\}$
8:    **end for**
9:    **for** $k \leftarrow 1$ to (# gradient steps) **do**
10:      Sample batch $\mathcal{B}_i \sim \mathcal{D}$
11:      Rearrange batch $\mathcal{B}_j = Rearrange(\mathcal{B}_i)$
12:      Sample a trainable Gaussian noise $\widetilde{\epsilon} = \mu_c + \sigma_1 f_\theta(\phi_{\bar{\omega}}(s_j), \phi_{\bar{\omega}}(s_j))$, where $\sigma_1 \sim \mathcal{N}(0, 1)$
13:      Train encoder $\mathbb{E}_{\mathcal{B}_i, \mathcal{B}_j}[\|\phi_\omega(s_i) - \phi_\omega(s_j)\|_2 - \mathcal{T}^{(T)}(s_i, s_j; \bar{\omega}, \theta)]$ with $\hat{\mathcal{P}}_\psi^\pi$ in Equation (10)
14:      Train the Actor-Critic: $\mathbb{E}_{\mathcal{B}_i}[\mathcal{L}^Q(\vartheta_{k=0,1}, \phi_\omega) + \mathcal{L}^\pi(\pi)]$ in Equation (22) and Equation (23)
15:      Train latent model $\mathbb{E}_{\mathcal{B}_i}\|\hat{\mathcal{P}}_\psi^\pi(\cdot|\bar{z}_t, a_t) - \bar{z}_{t+1}\|_2$ with frozen latent states
16:      Update target critics:$\mathbb{E}_{\mathcal{B}_i} Q_{\bar{\vartheta}_k}^{tgt} \leftarrow \tau_Q Q_{\vartheta_k} + (1 - \tau_Q)Q_{\bar{\vartheta}_k}^{tgt}, \forall k = 0, 1$
17:    **end for**
18: **end for**

---

# D EXPERIMENTAL SUPPLEMENT

## D.1 THE BEHAVIOR SIMILARITY UNDER ENCODER SPACES

To quantitatively analyze the accuracy of the converged SRL and DBC encoders in terms of behavioral similarity metrics, we calculate the encoding distances $d = \|\phi_\omega(s_i) - \phi_\omega(s_j)\|_2, (0 \le i, j < 500)$ of their pairwise combinations over 500 states and show the two pairs of states with the smallest encoding distance in Figure 8. It is worth noting that to avoid combinations with too close state transition steps, we set $|i - j| \ge 250$ to ensure the effectiveness of the encoding distance.

As depicted in Figure 8, on the left and right sides, we display the two closest groups of encoded distances for each DMC task under the SRL and DBC encoders, respectively. We can easily observe that the two sets of encoding states with the closest distance identified by the SRL encoder, also have very similar behavioral manifestation or task features intuitively. For example, in the `walker_run`, the differences between the states in each group are almost indistinguishable. In contrast, although the states classified by the DBC method have certain similarities, there are obvious differences in the detailed behaviors, so the effect is not good enough. In summary, the above results show that the weak bisimulation metric can effectively cluster similar states, and this ability actually requires SRL to accurately extract task-relevant features in the state, which strongly proves that SRL has powerful representation learning performance.

**SRL**                  **DBC**

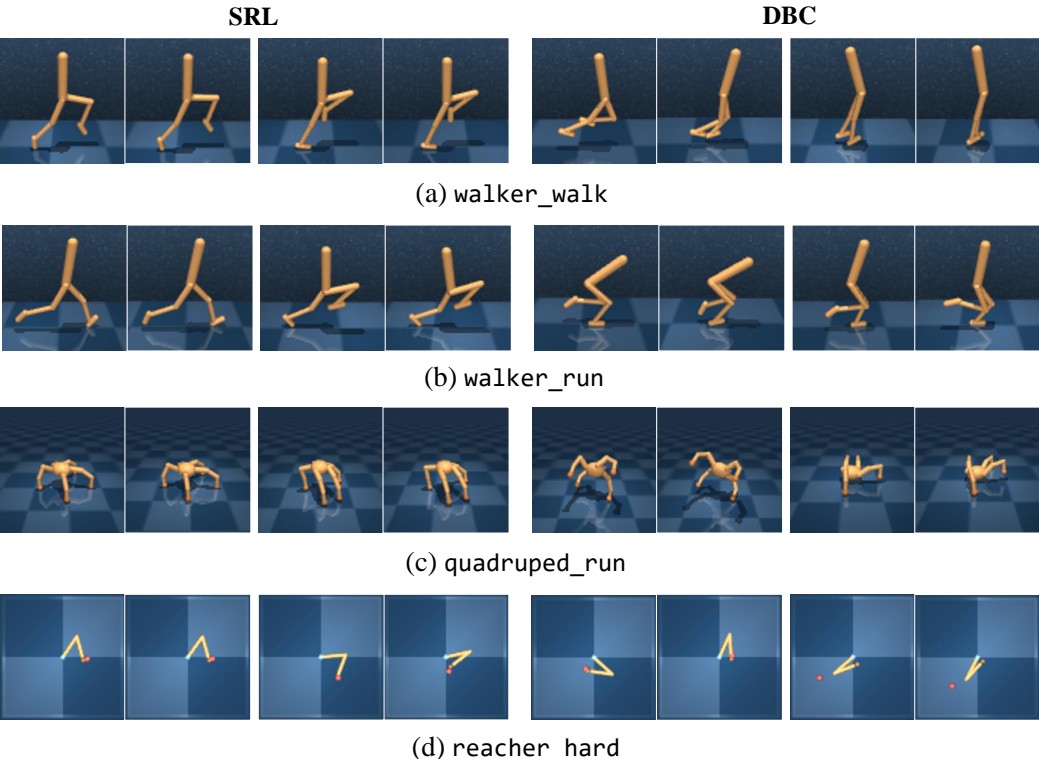

(a) `walker_walk`

(b) `walker_run`

(c) `quadruped_run`

(d) `reacher_hard`

Figure 8: Comparison of the behavioral similarity of state pairs under the latent space of SRL and DBC encoders. **Left**: SRL, **Right**: DBC.

## D.2 TASK-RELEVANT VISUALIZATIONS

As shown in Figure 9, we visualize the regions (green) of interest learned by convergent SRL and DBC encoders in the MetaWorld environment. Overall, we can observe that the features extracted by the SRL encoder (left side) are generally more complete and unambiguous. Interestingly, upon closer inspection, we notice that the SRL encoder may have also learned to recognize and emphasize potential safety boundaries, such as the operation boundaries on the desktop in the `box-close` task. In contrast, the encoding abilities of DBC (right) are bad, as they either only notice partial

components (e.g., cups and pallets), while failing to capture potential task components (e.g., buttons and target points), or result in disorganized and unclear visualized regions, such as the `box-close` task. More importantly, the task relationships between components, the abstract features of logic, and the scope of the components themselves seem difficult to represent. Due to the lack of an overall understanding of the task, it is difficult for the DBC to perform each necessary operation in an orderly manner and complete the task. To sum up, SRL can more systematically and clearly extract the entity features and abstract features required to support decision-making tasks compared to the baseline, and demonstrates strong representation learning capabilities in sparse reward environments.

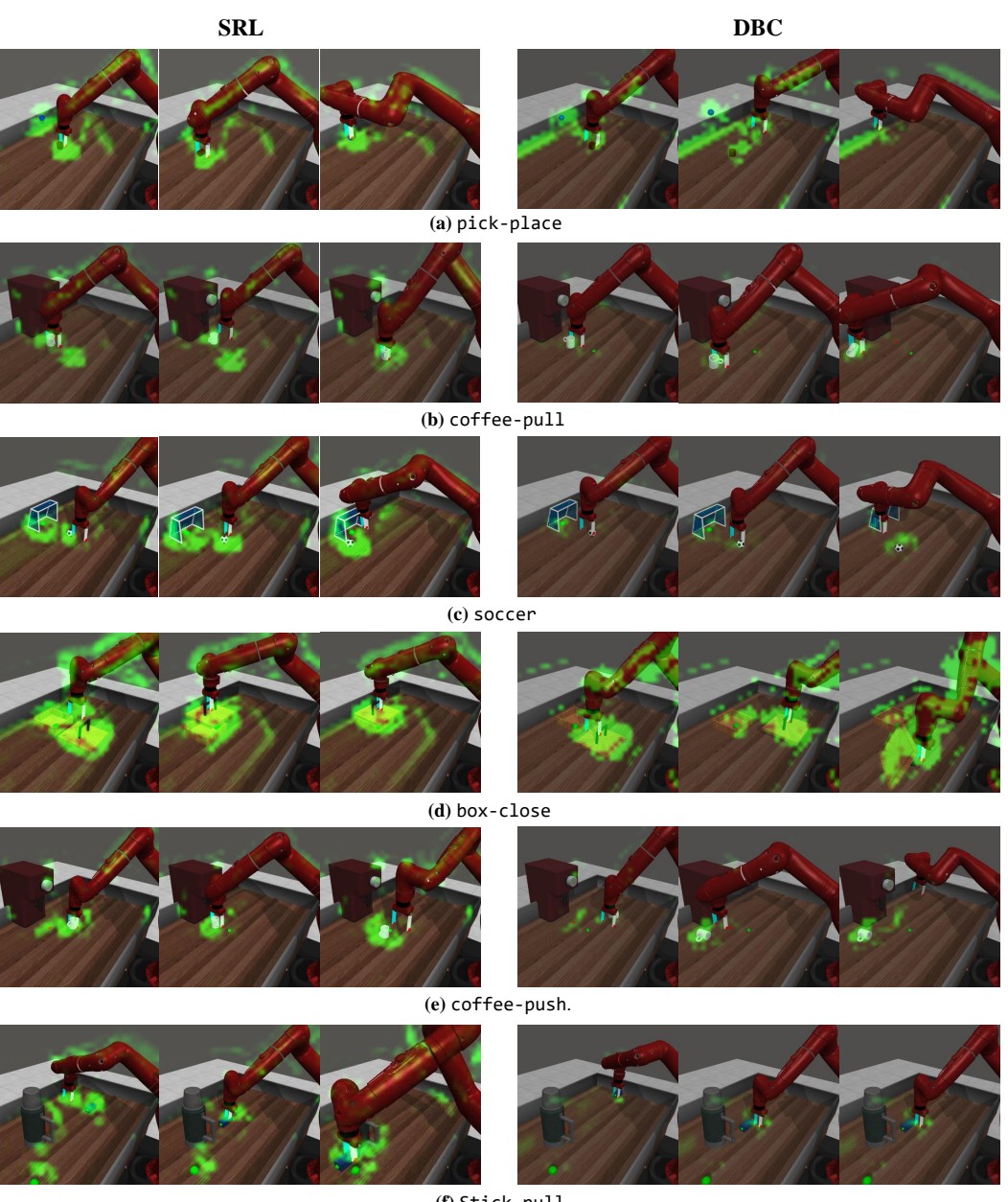

Figure 9: Visualization comparison between task-relevant features captured by the SRL and DBC encoders. **Left**: SRL, **Right**: DBC, with green heatmaps representing the task-relevant regions of interest as identified by the final convolutional layer of each encoder. Note that we employed consistent color parameters across both visualizations to ensure a fair comparison.

## D.3 REWARD COMPARISON

As shown in Figure 10 and Figure 11, we report the comparison curves of mean episode rewards between SRL and baselines in the MetaWorld and Adroit environments, respectively. The highest episode reward results are recorded in Table 4. We can observe that, despite the difficulty of the task settings, our method achieves the highest rewards across all tasks. As a key baseline for bisimulation-based methods, DBC performed poorly in almost all tasks, and in some cases, it struggled to achieve dense rewards within the limited training frames.

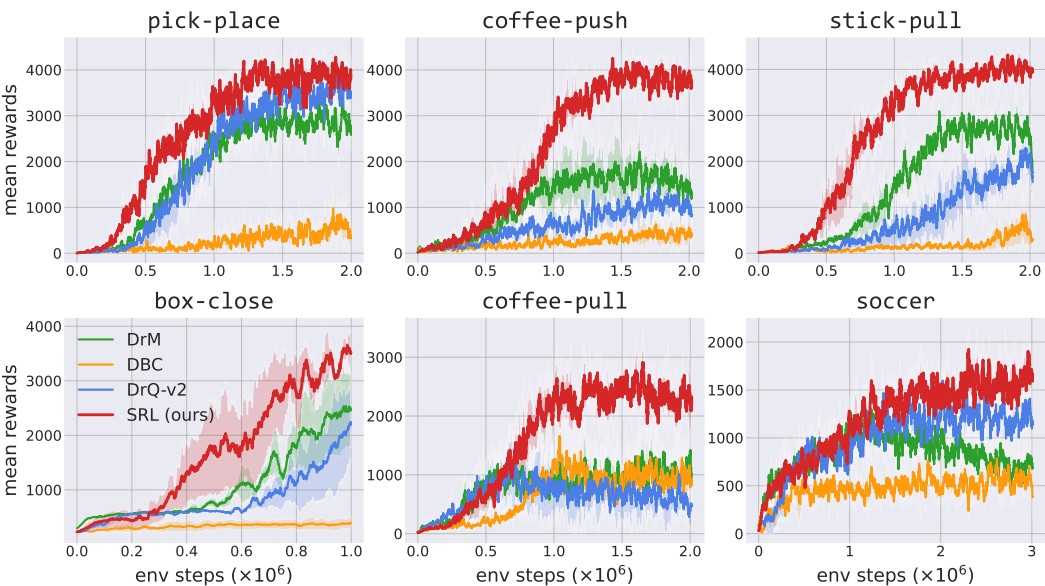

Figure 10: Learning curves of rewards for SRL and baselines on 6 complex tasks with sparse rewards in MetaWorld. Each curve represents the average of three random seeds, with the shaded regions indicating the standard deviation.

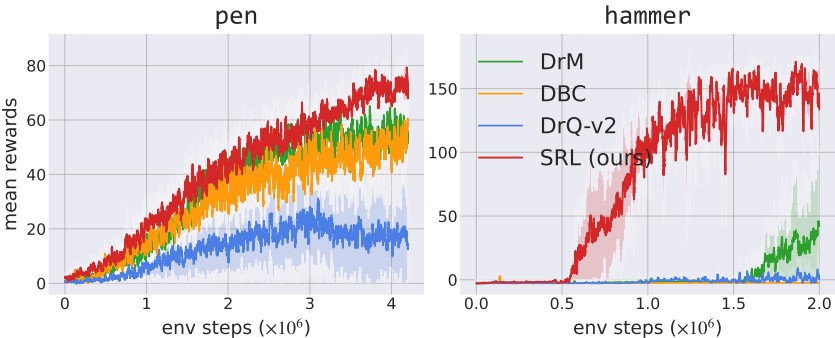

Figure 11: Learning curves of rewards for SRL and baselines on two complex tasks with sparse rewards in Adroit. Each curve represents the average of three random seeds, with the shaded regions indicating the standard deviation.

Table 4: Comparison results of the best mean episode reward on complex MetaWorld and Adroit tasks with sparse rewards.

| Methods | pick-place | coffee-push | stick-pull | box-close | coffee-pull | soccer | pen | hammer |
|---|---|---|---|---|---|---|---|---|
| DrM | 3345±80 | 2213±898 | 3091±168 | 2538±747 | 1424±253 | 1298±280 | 65.2±7.0 | 45.8±59.5 |
| DBC | 986±479 | 630±383 | 843±647 | 395±122 | 1665±58 | 784±241 | 60.6±3.7 | 2.7±0.0 |
| DrQ-v2 | 4061±76 | 1460±59 | 2294±421 | 2242±788 | 1225±572 | 1598±168 | 31.2±20.0 | 9.1±13.6 |
| SRL (ours) | **4281±194** | **4254±73** | **4319±110** | **3650±203** | **2916±559** | **1925±211** | **79.4±8.0** | **171.3±13.6** |

