# OpenReview forum: "Weak Bisimulation Metric-based Representations for Sparse-Reward Reinforcement Learning"
_ICLR.cc/2025/Conference — ICLR 2025 Conference Withdrawn Submission_

### Official Review · Reviewer_E3MW · 2024-10-23

**Soundness:** 3
**Presentation:** 3
**Contribution:** 3
**Rating:** 6
**Confidence:** 2

**Summary:**

The paper proposes a scalable representation learning approach (SRL) designed to improve the stability of representations under sparse-reward settings in reinforcement learning (RL). The core contribution is the introduction of a weak bisimulation metric, which relaxes traditional bisimulation metrics by eliminating the reliance on reward differences and incorporating a trainable Gaussian distribution. This relaxation is intended to address the challenges of representation collapse and degeneration that traditional bisimulation metrics face in sparse-reward environments. Additionally, the approach introduces continuous differences in the transition distribution to enhance task-relevant feature extraction. The proposed SRL is empirically validated on several sparse-reward RL benchmarks, such as DeepMind Control Suite, MetaWorld, and Adroit tasks, where it outperforms state-of-the-art methods.

**Strengths:**

The introduction of a weak bisimulation metric that relaxes strict assumptions while maintaining bisimulation's theoretical advantages is a novel and important step forward. The experiments are extensive, covering a variety of challenging tasks across multiple domains (DMControl, MetaWorld, Adroit), and the proposed SRL consistently outperforms baseline methods, including state-of-the-art approaches. The paper is generally well-structured, with a clear explanation of the problems with traditional bisimulation metrics and a well-motivated introduction of the weak bisimulation metric.

**Weaknesses:**

The paper assumes that Gaussian noise can effectively relax reward differences, but the implications of this assumption could vary across different environments. More detailed ablation studies that explore different noise distributions or parameterizations might provide further insights into the generalizability of the approach.

**Questions:**

Are there cases where the Gaussian assumption might not hold, and what alternatives would work better in those situations?

Could the performance of SRL be further improved by considering more complex forms of the transition distribution (e.g., non-Gaussian distributions), or is the flexibility provided by the Gaussian noise sufficient for most sparse-reward settings?

How does SRL compare with methods designed for dense-reward tasks? While the paper focuses on sparse-reward environments, could the approach offer any advantages in settings where rewards are more frequent?

---

> ### Author Response · Authors · 2024-11-26
>
> **Dear Reviewer:**
>
> We deeply appreciate your agreements and thoughtful suggestions. Our work is precisely aimed at addressing this challenge: mitigating the limited generalizability of standard bisimulation metrics in sparse reward settings while achieving significant improvements. Below are our point-by-point responses.
>
> **# Question 1**
>
> Thank you very much for the meaningful questions. First, our work focuses on the real-world sparse reward setting, where the reward collected on the policy distribution at each step of the environment transition is almost zero, i.e., the reward expectation
> $\mathbb E_{a_t\sim\pi}[r_{s_t}^{a_t}]$ is almost zero. To be conservative, we still assume that the reward expectation is no more than 0.1, so it is a weak assumption that fully satisfies the sparse reward setting, and even holds in environments with tiny dense reward settings (such as reacher-hard in DMControl). **Conversely, our assumption is not suitable for dense reward settings because their transition rewards are basically higher than 0.1. For such dense cases, the environment is already in a mild setting where standard bisimulation metrics or RAP methods (a bisimulation-based approach that weakens reward differences using reward variance) can perform effectively.** Our goal is to alleviate the limitations of Bisimulation metrics in sparse reward settings.
>
> **# Question 2**
>
> Thank you very much for the reviewer's detailed and very insightful questions. Currently, the flexibility provided by Gaussian noise is sufficient to meet the requirements of most environmental settings. In early investigations, **we happened to consider a uniform distribution scheme with the same mean, but we found that this distribution was not stable, i.e., it performed worse than SRL in the DMControl environment, slightly better than SRL in the MetaWorld environment, and even unstable across different tasks within the same environment. In contrast, using Gaussian noise has shown stable performance across various environments.** Overall, the design of the noise can significantly impact the performance of representation learning, and we are very pleased that the reviewer has raised such a forward-looking and insightful question.
>
> **# Question 3**
>
> Thank you very much to the reviewer for the constructive question. **In dense reward settings, SRL still achieves a slight advantage compared to methods based on dense rewards.** In fact, the experimental setups and results of this work can eliminate most of the reviewer's concerns. Specifically, our work deliberately includes experimental environments such as DMControl, MetaWorld, and Adroit, which correspond respectively to hard dense reward settings, sparse reward settings, and hard sparse reward settings (with large action space). Additionally, we include a dense reward-based baseline, $\pi$-bisimulation (DBC). For the comparative results in the DMControl tasks with dense rewards (Figure 4), SRL still demonstrates minor improvements in the simpler tasks, walker_walk and walker_run, and more significant improvements in the more challenging tasks, reacher_hard and quadruped_run. This indicates that SRL can maintain a slight advantage in generalized dense reward environments, and this advantage becomes more pronounced as the reward sparsity increases. It is worth noting that the four selected DMControl tasks actually feature weak dense reward signals, which are defined as hard tasks or informally categorized as sparse reward tasks in this work.
>
> **# Weakness**
>
> Thank you very much for your question. **You are correct that the implications of this assumption indeed varies across environments. Nevertheless, the Gaussian noise designed in this work is able to adapt to most environments, as explained in the responses to Question 1 and Question 2.** The exploration of the impact of different noise distributions is discussed in the response to Question 2, while the parameterization of reward differences or noise in the bsimulation metric is currently under investigation. We believe this is a very meaningful exploration.
>
> Thank you so much.

---

> > ### Comment · Reviewer_E3MW · 2024-11-26
> >
> > Thank you for the author's rebuttal. I maintain my current rating. I must also admit that I am not an expert in this field; therefore, my confidence level is 2.

---

> > > ### Author Response · Authors · 2024-11-29
> > >
> > > Dear Reviewer:
> > >
> > > Thank you for taking the time to review our work and your honest feedback. We appreciate your acknowledgment of our work and your efforts to evaluate our rebuttal.
> > >
> > > Best regards,
> > >
> > > All authors.

---

### Official Review · Reviewer_pua9 · 2024-11-03

**Soundness:** 1
**Presentation:** 1
**Contribution:** 2
**Rating:** 1
**Confidence:** 5

**Summary:**

The paper proposes a "reward-free" bisimulation-like metric to tackle the known representation collapse issue of bisimulation metrics in sparse reward environments. The expected-reward difference term in the $\pi$-bisimulation metric is replaced by a noise distribution and the transition distribution distance term is replaced via unrolling by a T-step discounted sum of future distances. Experiment results are shown over 3 pixel-based control suites against a few baseline algorithms.

**Strengths:**

- The empirical results presented in Sec. 5 seem encouraging.

**Weaknesses:**

1. The presentation seems too sloppy to me. The paper is littered with vague, unclear or incorrect statements that make it hard to read.
2. The introduction section is too dense and particularly lacking in good use of scientific language.

3. Some examples to the above two points:
- L13: "possess the superiority"
- L19 "intractable reward difference": Monte Carlo estimation makes this quite tractable.
- L23: "pure distribution internally": unclear what this means.
- L52: why "fundamentally"?
- L53: "equivalent" task-relevant features: equivalent how?
- L75: "certain favorable properties"?
- L78: "effective relaxation and strengthening in specific aspects"?
- L160: stacking frames does not lift partial observability
- L163: why is the reward a function of $\phi_\omega(s_t)$ and not $s_t$?
- L216: "In optional $d_\pi$": unclear what is being said here.
- L240: "But seriously,..."
- etc.

4. The motivation behind the particular alterations to the well-studied bisimulation formulation seem arbitrary and poorly motivated, sounding more like a marketing pitch than scientifically rigorous ideas. It is unclear to me how task relevance, which inherently depends on the reward function, is maintained when the reward difference term of bisimulation is replaced by some Gaussian noise arbitrarily.
5. The method is called "Scalable" Representation Learning, but it isn't clear to me how the scalability of the proposed approach is any different from prior work, e.g., MiCO by Castro et al. (2022).
6. The T-step unrolling of the transition distance seems like a hack and seems overly susceptible to high modelling errors due to compounding.
7. The identification of the representation collapse issue of $\pi$-bisimulation in sparse reward environments (the central focus of this paper) and Theorem 4.1 is incorrectly attributed to Liao et al. (2023), when in fact this was first studied by Kemertas et al. (2021); see their more general Lemma 2 with $c_R=1, c_T=\gamma$. The latter work is not mentioned until the end of Sec. 4.3 in Page 7.
8. Experiments do not compare to Kemertas et al. (2021), whose modifications of DBC (addition of embedding normalization, intrinsic rewards and inverse dynamics regularization) substantially improved performance, especially in sparse reward environments and would therefore comprise a stronger baseline.
9. L395: unclear why "challenging size of 2e5" is selected as the replay buffer size. This may be unfairly disadvantaging baseline methods.

**Questions:**

Why is the assumption A.1 not included in the main text, before the theorems that rely on them are presented?

---

> ### Author Response · Authors · 2024-11-18
>
> Dear reviewer:
>
> We sincerely appreciate the reviewers’ detailed feedback on our work and fully understand your concerns. We always attach great importance to the scientific nature of the methodology and the importance of clear and readable language in expressing the work. Also, we greatly respect the reviewers’ perspectives and are thankful for some constructive suggestions. However, we have different views on the rating and many detailed comments, and even think that the reviewers have some misunderstandings about this work. Below, we first provide a general response to these comments, followed by point-by-point replies. All our thorough responses and rebuttals aim to sincerely encourage you to reassess the significance of the problem, the methodology and experimental improvements presented in this work.
>
> **[General Response]**
>
> The reviewers highlighted numerous issues related to the clarity of our descriptions, suggesting that the work seemed sloppy and even hard to read. However, we feel it is necessary to emphasize that, upon a thorough review, most of the descriptions flagged by the reviewer as inaccurate or ambiguous can, in fact, be understood through context. Additionally, some of these descriptions are general terminologies, which are commonly used and reasonable in this field. Furthermore, three other reviewers unanimously considered the writing in our work to be clear, coherent, and even easy to understand and follow, indicating that the paper's presentation is effective for most readers. Nevertheless, we still incorporate the reviewers’ feedback to make comprehensive revisions to the manuscript.
>
> More importantly, we sincerely hope the reviewers focus more on the novelty of the SRL method itself and evaluate the effectiveness of its components in conjunction with the (ablation) experimental results, rather than making subjective assumptions beyond the scope of the method or the paper. For example: In **Weakness 4**, the reviewer claimed without sufficient evidence, that "our motivation sounds more like a marketing pitch than scientifically rigorous ideas" In fact, our experiments demonstrated that our metric constructed using $T-$step transition distribution differences with learnable Gaussian distributions significantly improves the agent’s decision-making performance in many complex visual environments. In **Weakness 6**, the reviewer dismissed the experimental results and discussion, suggesting that "The $T-$step unrolling of the transition distance seems like a hack and seems overly susceptible to high modelling errors due to compounding." However, we have fully discussed the model accuracy issue in the ablation study and showed that the advantage of the component outweighs the model error when $T=2$. For both examples, we disagree with the reviewer's view.
>
> Next, we will answer all the reviewers' questions and concerns one by one and state our different opinions. In particular, in Weakness 8, we further demonstrate the effectiveness of our method in a sparse reward environment compared to Kemertas et al. (2021) by supplementing necessary experiments.
>
>
> **[Point-by-point Response]**
>
> **#Re Weakness 1, Weakness 2, Weakness 3:**
>
> Thank you for the reviewer's detailed comments. Although the manuscript still has some shortcomings, we do not agree with the reviewer's extreme view on the "Representation" of this manuscript. At least from the consistent positive feedback from other reviewers on "Representation", we have reason to believe that the current manuscript can be effective for most readers. In fact, we have spent enough time polishing the abstract, introduction, methods and other chapters, and we will still make further revisions.
>
> **Explanation:**
> - L19 "intractable reward difference": Monte Carlo estimation makes this quite tractable. **Re:** Here we mean that the sparse reward setting leads to the invalidation of the internal reward difference of bisimulation metrics. We think that the effectiveness of Monte Carlo estimation is still based on the non-sparse reward setting of the environment, so it is still difficult to fundamentally solve the above problem.
> - L52: why "fundamentally"? **Re:** Its explanation starts from L53 "Distinct from..." In summary, bisimulation-based representation is different from most traditional representation learning techniques, and it theoretically has the potential to extract task-related information.
> - L216: "In optional $d_\pi$": unclear what is being said here. **Re:** "optional $d_\pi$" refers to the $π$-bisimulation-based DBC, MICO, and RAP metrics mentioned in the previous sentence.
> - L53: "equivalent" task-relevant features: equivalent how? **Re:** This equivalence is achieved by a bisimulation metric distance maintained by task elements such as rewards. The method aims to extract task-relevant features that are equivalent to this distance.
>
> -- Page 1--

---

> > ### Author Response · Authors · 2024-11-18
> >
> > - L160: stacking frames does not lift partial observability. **Re:** I completely agree with this statement. However, L160 does not express that “stacking frames can lift partial observability”. Our description is completely referred to Section 3.1 of the work [1]. If there is any misunderstanding, we can further modify it to avoid this misunderstanding.
> > Inaccurate description:
> > - L163: why is the reward a function of $\phi(s_t)$ and not $s_t$? **Re:** Thank you very much for the reviewer. This is a careless mistake, it is indeed $s_t$.
> > - L13: "possess the superiority".
> > - L75: "certain favorable properties".
> > - etc.
> >
> > **#Re Weakness 4:**
> >
> > (1) Although bisimulation formulas have been extensively studied and follow a fixed paradigm, there are still numerous challenges when applying them to deep reinforcement learning. To address this, many previous works have modified bisimulation formulas to varying extents. For instance, RAP introduces reward variance internally, and LIBERTY [2] incorporates an inverse dynamic differential term $\|\hat a_i(s_i,s_{i+1})- \hat a_j(s_j,s_{j+1})\|$, naming it the Inverse Dynamic Bisimulation Metric. In comparison, our work focuses on addressing representation collapse in sparse reward settings. Specifically, we propose constructing a simple and effective approximate bisimulation metric using a $T-$step dynamic transition difference term with trainable Gaussian distributions. A brief motivation can be found in #Re Weakness 5 (or see the Introduction, L68–L77).
> > (2) Task relevance is maintained through the $T-$step dynamic transition difference term. Its motivation stems from the fact that, during the whole state measurement process, utilizing the gap between consecutive $T-$step transition trajectories unrolling from two differ states as a metric will be beneficial to derive the dynamic task features guided by actions in the states. Additionally, the trainable Gaussian distributions serve as a mild buffer (values close to zero) to mitigate potential zero-distance metric issues, ensuring metric and representation stability.
> >
> > **#Re Weakness 5:**
> >
> > Our approach is based on the observation that sparse reward settings do not contribute anything to the reward difference and even harm the overall bisimulation metric. Hence, instead of weakly modifying the reward difference term (as RAP does, e.g., $\mathbb E_{a_i \sim \pi} \lbrack  | {r_{s_{i}}^{a_{i}} - r_{s_{j}}^{a_{j}}} |^2\rbrack  - Var \lbrack r_{s_i} \rbrack - Var \lbrack r_{s_j} \rbrack $, we replace the reward difference term with the Gaussian noise. This noise aims to ensure the weak relaxation of the metric and the representation by occupying a small space, ensuring their stability. Further, we accordingly use $T-$step transition difference to strengthen state similarity metrics. We describe representation learning based on this combination of relaxation and strengthening operations as a scalable representation learning method. For MICo, the problem it solves is completely different from SRL. MICo aims to develop the MICo operator to improve the Wasserstein distance with high computational complexity.
> >
> > **#Re Weakness 6:**
> >
> > We fully understand your concern. However, as an auxiliary task, $T-$step dynamic transitions can still improve the main task when a balance is achieved, such as the previous Latent Overshooting [3]. In fact, our ablation study verifies that when $T=2$, the performance improvement brought by this structure is greater than the consumption of model accuracy, and the overall improvement is positive. It is worth noting that we have actually discussed and concluded this issue in Section 5.3, L523.
> >
> > **#Re Weakness 7:**
> >
> > In our original manuscript, we did not attribute the identification of representation collapse to Liao et al. (2023) [4]. We are fully aware that the foundational work is by Kemertas et al. (2021) [5], which we have discussed and cited earlier in the manuscript. Nevertheless, Liao et al. (2023) made some specific extensions to this problem, and their Lemma 3.1 (i.e., our Theorem 4.1) is conducive to the unfolding of our Section 4, so we should cite Liao et al. (2023) here.
> >
> > -- Page 2 --

---

> > > ### Author Response · Authors · 2024-11-18
> > >
> > > **#Re Weakness 8:**
> > >
> > > Thank you very much for the reviewer's constructive suggestions on the experiments. We have supplemented the comparative experiments with the method of Kemertas et al. (2021) [5], as shown in Table 1 below (the evaluation curve can be seen in Figure 12 in the appendix file). Specifically, we insert SRL into their framework to maintain completely consistent parameter and environment settings. Furthermore, we selected "ContinuousCartpole-v0" and "SparsePendulum-v0" with reward sparse modification as test tasks, and completed three sets of comparative experiments with the $N_m$={1,2,3} distractor noise dimensions respectively. Overall, we generally outperform them in terms of the best scores (Table 1). P
> > > Particularly, as shown in Figure 12, SRL converges significantly faster than theirs, yielding a non-negligible improvement. Additionally, following the original work, embedding normalization is critical for performance. We also showcase their method’s performance under a “without normalization” i.e., “no norm” setting. In contrast, SRL achieves the best performance even without embedding normalization. Finally, it is worth noting that we did not consider the "MountainCarContinuous-v0" task in the original work because it is difficult to reproduce.
> > >
> > > **Table 1: Comparison of the highest scores of SRL and Kemertas et al. (2021) method. Each result is the average of 10 seeds.**
> > >
> > > | Method                     | Sparse Cartpole ($N_m$=1) | Sparse Cartpole  ($N_m$=2) | Sparse Cartpole ($N_m$=3)  | Sparse Pendulum ($N_m$=1) | Sparse Pendulum ($N_m$=2) | Sparse Pendulum ($N_m$=3) |
> > > |----------------------------|------------------------|------------------------|------------------------|------------------------|------------------------|------------------------|
> > > | Kemertas et al.            | 200.0 ± 0.0           | **200.0 ± 0.0**            | **198.40 ± 2.99**          | 153.37 ± 41.67         | 164.70 ± 5.85          | 147.25 ± 51.51         |
> > > | Kemertas et al. (no norm)  | 69.91 ± 46.82         | 38.28 ± 4.19           | 31.81 ± 25.03          | 159.81 ± 19.23         | 163.22 ± 7.39          | 90.38 ± 80.48          |
> > > | SRL (Ours)                       | **200.0 ± 0.0**           | 197.97 ± 3.00          | 194.84 ± 4.03          | **170.87 ± 3.36**          | **171.37 ± 2.09**          | **171.54 ± 2.47**          |
> > >
> > > **#Re Weakness 9:**
> > > We appreciate your suggestion, and we will use additional ablation experiments to dispel ambiguity. In fact, our initial experiments show that the method results with a size of 1e6 are still significantly better than baselines. Therefore, this work chooses a replay buffer size of 2e5 mainly to reduce the reward signal of the buffer to further challenge the performance of different methods, especially observing the stable performance of SRL; on the other hand, it saves memory consumption.
> > >
> > > **#Re Question 1**
> > > We can move it to the main text according to the reviewer's suggestion. Before that, we mainly considered that Assumption A.1 was too long and a weak assumption, and was only used in the internal proof process of Theorem 4.4 and Theorem 4.5, so we chose to put it before the appendix proof. In fact, we discussed this assumption in the main text, L294.
> > >
> > >
> > > Thank you.
> > >
> > >
> > > **[Reference]**
> > >
> > > [1] Liu Q, Zhou Q, Yang R, et al. Robust representation learning by clustering with bisimulation metrics for visual reinforcement learning with distractions. AAAI, 2023.
> > >
> > > [2] Wang Y, Yang M, Dong R, et al. Efficient potential-based exploration in reinforcement learning using inverse dynamic bisimulation metric. NeurIPS, 2024.
> > >
> > > [3] Hafner D, Lillicrap T, Fischer I, et al. Learning latent dynamics for planning from pixels. ICML, PMLR, 2019.
> > >
> > > [4] Liao W, Zhang Z, Yu Y. Policy-independent behavioral metric-based representation for deep reinforcement learning. AAAI, 2023.
> > >
> > > [5] Kemertas M, Aumentado-Armstrong T. Towards robust bisimulation metric learning. NeurIPS, 2021.
> > >
> > > -- Page 3 --

---

> ### Author Response · Authors · 2024-11-24
>
> Dear Reviewer:
>
> We deeply appreciate the time and efforts. Given the importance of clarifying potential misunderstandings and addressing the reviewer’s concerns, we wanted to check if there might be an update regarding the feedback. However, it has been almost a week since we submitted our rebuttal, and we have not received any further responses.  We believe that the rebuttal stage is an important part of the ICLR review process, and thus we have addressed the reviewers' comments fully and carefully. We also sincerely hope that reviewer will not hesitate to reply to us and update your views.
>
> Thank you.

---

> ### Author Response · Authors · 2024-11-29
>
> Dear Reviewer,
>
> Thank you once again for taking the time to review our paper and for your valuable insights. We understand that the review process can be time-consuming, and we greatly appreciate your effort and expertise.
>
> We are kindly following up regarding the rebuttal phase. As we have provided our responses to your feedback earlier, we would sincerely appreciate it if you could take a moment to review them and share any additional thoughts or updates.
>
> Thank you very much for your time and consideration.
>
> Best regards,
>
> All authors.

---

### Official Review · Reviewer_fcZ3 · 2024-11-04

**Soundness:** 2
**Presentation:** 3
**Contribution:** 2
**Rating:** 3
**Confidence:** 4

**Summary:**

This paper investigates the problem of representation learning in deep reinforcement learning. It points out the fundamental challenges of previous (approximations of) bi-simulation metrics. It proposes SRL, which relaxes bi-simulation metric, aiming at solving the intractable reward difference and collapse in the sparse reward setting. Furthermore, it considers continuous differences over the transition distribution to tighten the metric. Finally, experiments are conducted to show the advantage of this algorithm.

**Strengths:**

1. Most part of this paper is easy to read and follow.
2. The experiment results shows that the resulting algorithm is consistently better than baselines.

**Weaknesses:**

1. The writing about some concepts can be improved. While I understand bi-simulation is not new. Detailed definitions of relevant concepts should be provided, at least in the appendix. For example one would not be able to understand Theorem 3.1 if one does not read other papers. Besides, definition 4.2 is also not strict as it contains random variable in the definition of a deterministic map by default. This makes subsequent statements unclear as well.
2. If I am understanding correctly, this paper contains overstatements on theorems. See question 2.
3. Definition 4.2 seems unnatural to me, see question 3. More explanations should be provided

Typos:
1. In definition 4.2, the minus sign should be plus.

**Questions:**

1. What is the formal definition of $\mathbb{M}$ in Theorem 3.1? What is the definition of $\pi$-simulation and what is a least fixed point?
2. Can you explain the logic in line 234 and 235? Theorem 4.1 only reasons about the sup over all states rather than some specific state pair.
3. By introducing a Guassian random variable in the definition 4.2, the distance can now be negative. Besides, when $s_i=s_j$, this metric does not yield zero, not even on average. Why does it make sense
4. I am assuming scalability is the ability to remain effective when performing on more complex tasks. In what sense is SRL scalable?

---

> ### Author Response · Authors · 2024-11-21
>
> Dear Reviewer：
>
> We are very grateful to the reviewer for the careful review of our work and the useful suggestions. Below is our detailed point-to-point response to the reviewer's specific questions, hoping to do our best to eliminate the reviewer's confusion. Finally, we sincerely hope that the reviewer can evaluate our work again.
>
> **[Point-by-point Response]**
>
> **#Re Question 1:**
>
> **(1)** $\mathbb M$ is defined as the set of pseudometrics $d^\pi:\mathcal{S}\times\mathcal{S}\rightarrow\left[0,\infty\right)$ on the state space $\mathcal{S}$, i.e. $d^\pi\in\mathbb M$, which allows providing a metric value $d^\pi(s_i,s_j)$ for each pair of states $(s_i,s_j)\in \mathcal{S}\times\mathcal{S}$.  **(2)** Perhaps you mean $\pi$-bisimulation. It is defined as the sum of the immediate reward difference and the state transition distribution difference under a given $\pi$, see **Theorem 3.1** for details. Instead of considering the similarity of states for all possible policies (differ to bisimulation), $\pi$-bisimulation refers to the behavior similarity between two states under a specific policy $\pi$.   **(3)** It is actually difficult to derive an explicit expression for the least fixed point. In fact, we only need its existence. Once its existence is proved, it shows that the metric $d^\pi$ will iteratively converge to a fixed measure over the state pair, laying a convergence foundation for representation learning constructed with this metric $d^\pi$.
>
> **# Re Question 2:**
>
> We understand the reviewer's concern, but we did not actually overstatement **Theorem 4.1**. The explanation of L234-L235 is as follows: For the mildly dense reward setting, if the policy performs very poorly in the early training, the collected rewards are zero everywhere, which leads to the internal $\left|r_{s_i}^{a_i}-r_{s_j}^{a_j}\right|=0$. This will further produce a degenerate solution, i.e., $\text{diam}(\mathcal{S},d^\pi)=0$ (or $d^\pi=0$). However, in practical sparse reward settings, the above extreme case will become very common, making the supremum of the reward differential is basically zero, i.e., $ sup_{s_i,s_j \in \mathcal{S}}\left|r_{s_i}^{a_i}-r_{s_j}^{a_j}\right|=0$, which will continuously destroy the correct convergence of $d^\pi$. In fact, the Metaworld and Adroit environments verified in our work are both sparse reward settings, and do not exaggerate the existence of such extreme cases and the extension of the **Theorem 4.1**. Nevertheless, we will further improve the description of this part.
>
> **#Re Question 3:**
>
> This is a representative and meaningful question. We believe the following explanation can solve your problems and help you understand our motivation in depth. We will start by introducing the problem our work solves and our motivation, and then answer the questions step by step.
>
> * **The problem resolved**
>
> The $\pi$-bisimulation metric $d^\pi(s_i,s_j) \boldsymbol{(\geq0)}$ aims to measure the behavioral similarity between states by the reward and the next state distances, which is widely used in visual reinforcement learning to construct an auxiliary representation learning. During the optimization process, $\pi$-bisimulation metric iteratively updates the $d^\pi(s_i,s_j)$ until it converges to the fixed point $d_B^\pi(s_i,s_j)$, at which time the similarity distance between any state pairs is stabilized, allowing the encoder to learn a stable representation space. However, the reward difference term $\left|r_{s_i}^{a_i}-r_{s_j}^{a_j}\right|$ inside the $d^\pi(s_i,s_j)$ is very susceptible to zero reward signals, which ultimately interferes with the convergence direction of $d^\pi$ and produces a degenerate solution $\text{diam}(\mathcal{S},d^\pi)=0$. This will directly lead to the collapse of the learned representation space. This challenge was first addressed by Kemertas et al. (2021) and further studied by Liao et al. (2023).
>
> * **Motivation and Response Q3-1**
>
> In summary, considering that the internal reward difference not only does not contribute but even harms the representation under the sparse reward setting, this work attempts to replace it with a trainable Gaussian reward distribution. The introduced Gaussian random variable $\epsilon\geq0$ (where $\boldsymbol{\mu>0}$) with clipping operations can be considered as a tiny placeholder space inside the $d^\pi$. It can commonly ensure the $d^\pi(s_i,s_j)>\epsilon$ in the worst case of continuous zero reward signal, avoiding the above-mentioned representation collapse problem caused by $\text{diam}(\mathcal{S},d^\pi)=0$. It is worth noting that since $\epsilon$ is sampled from a trainable parameterized Gaussian distribution, it will also be adaptively adjusted with the transition distribution difference term, thereby maintaining the stability of the metric $d^\pi(s_i,s_j)$ and representations.

---

> > ### Author Response · Authors · 2024-11-21
> >
> > * **Response Q3-2**
> >
> > For the special case of $s_i=s_j$, in fact, their behavioral similarity $d^\pi(s_i,s_j)$ is usually greater than zero, i.e., not completely similar. This is because $d^\pi(s_i,s_j)=0$, i.e., the premise for a state pair to be a mutual simulation relationship is that the two have the same reward and state transition distribution. Thus, in this case $d^\pi(s_i,s_j)>0$  which is in the right direction. We use this problem to illustrate the strength of our approach: Although in sparse reward environments there is a unavailable true reward for each state transition, the actual reward signal generated by the environment is often zero. This leads to the traditional bisimulation metric easily misestimating that the state behavior is similar due to the environmental zero reward signal when dealing with the case $s_i=s_j$, which will further lead to the collapse of the representation space. In contrast, we use a small positive random variable to ensure the non-zero property of $d^\pi(s_i,s_j)$. As the initial motivation, our experiments show that this can at least ensure the effectiveness of the overall behavior similarity metric, although it is still difficult to recover the invalidation of the reward item.
> >
> > **#Re Question 4:**
> >
> > Thank you very much for your suggestion. Initially, "scalability" in this work refers to the scalability of the method framework. Concretely, on the one hand, we use trainable Gaussian noise to relax the reward term inside the $\pi$-bisimulation metric; On the other hand, to cope with the above relaxation, we take multi-step transition operations on the transition difference term to strengthen the weakened metric, improving the accuracy of the state similarity metric $d^\pi(s_i,s_j)$ by calculating the latent trajectory distance of the state pair. We describe this two-stage relaxation and strengthening as a scalable representation learning. Therefore, this may be different from what the reviewer thinks, but we will carefully consider the accuracy of the "scalability" description.
> >
> > Thank you.

---

> ### Author Response · Authors · 2024-11-24
> **Follow-Up on Rebuttal Discussion**
>
> Dear Reviewer,
>
> Thank you for your detailed review and valuable suggestions. We understand that you may have a busy schedule, but we kindly hope you could take a moment to provide any replies or possible updates regarding our rebuttal.
>
> We greatly appreciate your time and consideration, and we look forward to your feedback.
>
>
> Best regards,
>
> All authors

---

> ### Author Response · Authors · 2024-11-29
>
> Dear Reviewer,
>
> Thank you once again for taking the time to review our paper and for your valuable insights. We understand that the review process can be time-consuming, and we greatly appreciate your effort and expertise.
>
> We are kindly following up regarding the rebuttal phase. As we have provided our responses to your feedback earlier, we would sincerely appreciate it if you could take a moment to review them and share any additional thoughts or updates.
>
> Thank you very much for your time and consideration.
>
> Best regards,
>
> All authors.

---

### Official Review · Reviewer_RnBn · 2024-11-04

**Soundness:** 2
**Presentation:** 2
**Contribution:** 2
**Rating:** 5
**Confidence:** 4

**Summary:**

This paper proposes a weak bisimulation metric in sparse reward settings. Compared to the previously strict bisimulation metric methods, the weak bisimulation metric introduces two primary enhancements: (1) it relaxes the reward difference term through a trainable Gaussian distribution, alleviating potential representation collapse caused by sparse rewards; (2) it strengthens the extraction of equivalent task features by accumulating state transition distribution differences accordingly. Experimental results on DMC, Meta-World, and Adroit demonstrate that, the weak bisimulation metric indeed improves performance in sparse reward tasks.

**Strengths:**

The introduction presents a compelling motivation and provides a clear background explanation.

The proposed weak bisimulation metric is a straightforward yet effective approach, and it is easily adaptable to other bisimulation-based algorithms.

**Weaknesses:**

**1. Insufficient Coverage of Related Work.**

The current manuscript lacks adequate discussion of prior work related to bisimulation representation collapse in sparse reward tasks. Notable approaches, such as constructing bisimulation metrics through intrinsic rewards and inverse dynamics [1] or adopting action-based bisimulation to eliminate reward dependency [2], are absent. To strengthen the comparison, I recommend that the authors clarify how their proposed method differs from or improves upon these existing techniques. Specifically, they could discuss the relative advantages of their approach compared to the intrinsic reward method in [1] and the action-based method in [2].

**2. Errors and Ambiguities.**

This paper contains several errors in expression and ambiguities that could affect clarity and accuracy. For example, “−” is used instead of “+” in Eq. (5) and Eq. (17). Additionally, in line 402, the statement "We choose complex *walker_run*, *walker_walk*, *reacher_hard*, and *quadruped_run* tasks with sparse reward properties" incorrectly categorizes *Walker* and *Quadruped* as sparse reward tasks, which are typically considered dense reward tasks as per [3]. To improve clarity, I suggest that the authors carefully review these equations and statements. It may also be helpful for them to provide clarification if their categorization of tasks differs from the standard definitions in [3], to ensure readers understand any intentional deviations from established terminology.

**3. Experimental Design and Comparison Issues.**

The experimental setup has several issues:

(1)The selection of comparative algorithms lacks sufficient representativeness. DrQ-v2 and DrM, while valuable, are not RL algorithms specifically designed to use bisimulation for extracting task-relevant representations. It would strengthen the study to include additional bisimulation-focused algorithms, such as MICo [4] and RAP [5], as well as other methods addressing bisimulation in sparse reward tasks (e.g., [1] and [2]). I suggest that the authors clarify their rationale for the current baseline selection and consider a broader comparison with bisimulation-based approaches to provide a more comprehensive evaluation.

(2)In Figure 4, It is unfair to compare SRL (build upon DrQ-v2) proposed in this paper with DBC, as DrQ-v2 demonstrates significantly superior performance than DBC.

(3)In the selection of experimental tasks, it is necessary to include experiments with a noisy background. Additionally, a more in-depth qualitative analysis of the relationship between introduced redundant information and interference from noisy backgrounds is required.

(4)This paper would benefit from ablation experiments to analyze the contributions of individual modules. Specifically, an ablation study comparing the model's performance with and without the Gaussian distribution term in the weak bisimulation metric could clarify its impact and importance within the framework.

**4. Reliance on Assumption A.1.**

The paper’s theoretical results rely heavily on Assumption A.1, which presumes that the sparse-reward expectation remains less than or equal to a sufficiently small constant $C_1$. However, the assumption lacks precise quantification, particularly regarding what constitutes a “sufficiently small” constant, potentially limiting the applicability of the results. I recommend that the authors either provide empirical evidence from their experiments to support the validity of this assumption or discuss the possible implications if this assumption does not hold in certain environments.

[1]Kemertas M, Aumentado-Armstrong T. Towards robust bisimulation metric learning. NeurIPS, 2021.

[2]Rudolph M, Chuck C, Black K, et al. Learning Action-based Representations Using Invariance[C]. RLC, 2024.

[3]Tassa Y, Doron Y, Muldal A, et al. Deepmind control suite[J]. arXiv preprint arXiv:1801.00690, 2018.

[4]Castro P S, Kastner T, Panangaden P, et al. MICo: Improved representations via sampling-based state similarity for Markov decision processes[C]. NeurIPS, 2021.

[5]Chen J, Pan S. Learning representations via a robust behavioral metric for deep reinforcement learning[J]. NeurIPS, 2022.

**Questions:**

What is the specific meaning of the mean $\mu_c$ in the Eq.(5)? It appears that the mean $\mu_c$ is merely a constant without practical significance. And how can it be ensured that the weak bisimulation metric effectively extracts task-relevant features?

---

> ### Author Response · Authors · 2024-11-21
>
> Dear Reviewer：
>
> Thank you very much for the reviewer's careful review and valuable time. We think that many of your comments (such as Weakness 1 and Weakness 2) have been very helpful in the better presentation of our manuscript. To alleviate the reviewer's concerns about the baseline and noise experiments in bisimulation-based representation learning, we added the comparison results of the new baseline RAP [1] in the MetaWorld sparse-reward environment, and the comparison results of the SRL method with Kemertas et al. [2] in a noisy Gym environment. Finally, we sincerely hope that if we have addressed the reviewer' concerns, the reviewer will be able to evaluate our work again. The following is a detailed point-to-point response:
>
> **#Re Weakness 3-1**
>
> To mitigate the poor performance of Bisimulation representations in sparse reward settings, RAP introduces reward variance, i.e., $\mathbb E_{a_i \sim \pi} \lbrack  | {r_{s_{i}}^{a_{i}} - r_{s_{j}}^{a_{j}}} - Var(r_{s_i}) - Var(r_{s_j}) \rbrack $, into the reward difference term within the metric, thereby mitigating the unstable metrics and representation learning process caused by rewards. For effective comparison, we tested the RAP method on 6 tasks in the MetaWorld environment with a standard sparse reward setting. The following are the experimental comparison results:
>
> **Table 1: Comparison of the highest scores of SRL and RAP. Each result is the average of three seeds.**
>
> | Method   | Pick-place (%)       | Coffee-push (%)      | Stick-pull (%)       | Box-close (%)        | Coffee-pull (%)      | Soccer (%)           |
> |----------|----------------------|----------------------|----------------------|----------------------|----------------------|----------------------|
> | DrM      | 86.4±2.9            | 73.5±14.0           | 27.8±35.8           | 95.7±2.5            | 53.8±7.2            | 48.8±2.0            |
> | RAP      | 92.6±2.5            | 91.5±4.6            | 86.7±4.4            | 54.9±39.8           | 79.8±2.5            | **78.8±6.4**            |
> | DBC      | 25.3±10.9           | 44.7±5.2            | 16.1±13.6           | 0.0±0.0             | 49.2±11.3           | 29.6±8.5            |
> | DrQ-v2   | 98.7±1.9            | 63.0±6.1            | 2.5±3.5            | 97.5±2.0            | 60.8±5.1            | 52.5±4.4            |
> | SRL (ours) | **99.3±0.9**        | **98.0±0.0**        | **92.9±4.6**        | **98.0±0.0**        | **83.1±6.2**        | 67.9±4.8        |
>
> Overall, our SRL achieves the best performance on the best episode scores in most tasks. Moreover, SRL actually converges significantly faster than RAP on the box-close, stick-pull, and pick-place tasks.
>
> **#Re Weakness 3-2**
>
> Thank you for your comments. We always attach great importance to the fairness of comparison. Among the methods involved in our experiments, they are all built on DrQ-v2 (such as DBC, which is actually DrQ-v2+DBC) and use consistent hyperparameters.
>
> **#Re Weakness 3-3**
>
> We have supplemented the comparative experiments with Kemertas et al. (2021), as shown in Table 2 below (the evaluation curve can be seen in Figure 12 in the appendix file). Specifically, we insert SRL into their framework to maintain completely consistent parameter and environment settings. Furthermore, we selected "ContinuousCartpole-v0" and "SparsePendulum-v0" with reward sparse modification as test tasks, and completed three sets of comparative experiments with the $N_m$={1,2,3} distractor noise dimensions respectively. Overall, we generally outperform them in terms of the best scores (Table 2). Particularly, as shown in Figure 12, SRL converges significantly faster than theirs, yielding a non-negligible improvement. Additionally, following the original work, embedding normalization is critical for performance. We also showcase their method’s performance under a “without normalization” i.e., “no norm” setting.
>
> **Table 2: Comparison of the highest scores of SRL and Kemertas et al. (2021) method. Each result is the average of 10 seeds.**
>
> | Method                     | Sparse Cartpole ($N_m$=1) | Sparse Cartpole  ($N_m$=2) | Sparse Cartpole ($N_m$=3)  | Sparse Pendulum ($N_m$=1) | Sparse Pendulum ($N_m$=2) | Sparse Pendulum ($N_m$=3) |
> |----------------------------|------------------------|------------------------|------------------------|------------------------|------------------------|------------------------|
> | Kemertas et al.            | 200.0 ± 0.0           | **200.0 ± 0.0**            | **198.40 ± 2.99**          | 153.37 ± 41.67         | 164.70 ± 5.85          | 147.25 ± 51.51         |
> | Kemertas et al. (no norm)  | 69.91 ± 46.82         | 38.28 ± 4.19           | 31.81 ± 25.03          | 159.81 ± 19.23         | 163.22 ± 7.39          | 90.38 ± 80.48          |
> | Ours                       | **200.0 ± 0.0**           | 197.97 ± 3.00          | 194.84 ± 4.03          | **170.87 ± 3.36**          | **171.37 ± 2.09**          | **171.54 ± 2.47**          |

---

> ### Author Response · Authors · 2024-11-21
>
> **#Re Weakness 3-4**
>
> We understand the reviewer's concerns very well. In fact, the ablation study in Section 5.3 can alleviate most of the reviewer's concerns. In Section 5.3, we perform ablation experiments on $T$-step transition differences inside the weak bisimulation metric for $T$={0,1,2,3}. For $T=0$, since our metric containing only noise terms no longer holds, it is regarded as a normal DrQ-v2 method. Therefore, we analyze the effectiveness of the weak bisimulation metric and the effectiveness of $T$-step transition differences through $T$={0,1} and $T$={1,2,3,} comparison results.
>
> **#Re Weakness 4 and Question 1:**
>
> Thank you very much for your constructive suggestions. In fact, our assumption A.1 is a very weak assumption about expected immediate rewards that holds almost perfectly in the standard sparse reward setting, or can even be removed. Therefore, this assumption is mainly to take into account some difficult tasks with small rewards in DMControl (such as reacher-hard, etc.). We apologize for the question 1, $\mu_c$ is a formal description of the mean of reward expectation, which is indeed a redundant symbol that can be covered by the $C_1$ parameter. We will modify this symbol later.
>
> [1]Chen J, Pan S. Learning representations via a robust behavioral metric for deep reinforcement learning[J]. NeurIPS, 2022.
> [2]Kemertas M, Aumentado-Armstrong T. Towards robust bisimulation metric learning. NeurIPS, 2021.
>
> Thank you very much.

---

> ### Author Response · Authors · 2024-11-24
> **Follow-Up on Rebuttal Discussion**
>
> Dear Reviewer,
>
> Thank you for your detailed review and valuable suggestions. We understand that you may have a busy schedule, but we kindly hope you could take a moment to provide any replies or possible updates regarding our rebuttal.
>
> We greatly appreciate your time and consideration, and we look forward to your feedback.
>
> Best regards,
> All authors

---

> ### Author Response · Authors · 2024-11-29
>
> Dear Reviewer,
>
> Thank you once again for taking the time to review our paper and for your valuable insights. We understand that the review process can be time-consuming, and we greatly appreciate your effort and expertise.
>
> We are kindly following up regarding the rebuttal phase. As we have provided our responses to your feedback earlier, we would sincerely appreciate it if you could take a moment to review them and share any additional thoughts or updates.
>
> Thank you very much for your time and consideration.
>
> Best regards,
>
> All authors.

---

### Note · Authors · 2024-12-03

I have read and agree with the venue's withdrawal policy on behalf of myself and my co-authors.